# Monkey plays Pac-Man with compositional strategies and hierarchical decision-making

Qianli Yang[1], Zhongqiao Lin[1], Wenyi Zhang[1,2], Jianshu Li[1], Xiyuan Chen[1,2], Jiaqi Zhang[3], Tianming Yang[1,4]*

[1]Institute of Neuroscience, Key Laboratory of Primate Neurobiology, Center for Excellence in Brain Science and Intelligence Technology, Chinese Academy of Sciences, Shanghai, China; [2]University of Chinese Academy of Sciences, Beijing, China; [3]Brown University, Providence, United States; [4]Shanghai Center for Brain Science and Brain-Inspired Intelligence Technology, Shanghai, China

**Abstract** Humans can often handle daunting tasks with ease by developing a set of strategies to reduce decision-making into simpler problems. The ability to use heuristic strategies demands an advanced level of intelligence and has not been demonstrated in animals. Here, we trained macaque monkeys to play the classic video game Pac-Man. The monkeys' decision-making may be described with a strategy-based hierarchical decision-making model with over 90% accuracy. The model reveals that the monkeys adopted the take-the-best heuristic by using one dominating strategy for their decision-making at a time and formed compound strategies by assembling the basis strategies to handle particular game situations. With the model, the computationally complex but fully quantifiable Pac-Man behavior paradigm provides a new approach to understanding animals' advanced cognition.

*For correspondence:
tyang@ion.ac.cn

Competing interest: The authors declare that no competing interests exist.

## Editor's evaluation

Dr. Yang and colleagues trained nonhuman primates (rhesus monkeys) to play a semi-controlled version of the video game Pac-Man. This novel experimental paradigm allowed the authors to analyze and model the kinds of heuristic behavioral strategies monkeys use to solve relatively complex problems. The results provide insight into higher cognition in primates.

## Introduction

Our lives are full of ambitious goals to achieve. Often, the goals we set out to accomplish are complex. For it to be acquiring a life with financial stability or winning the heart of your love of life, these ambitious goals are often beyond the reach of any straightforward decision-making tactics. They, however, may be approached with a specific and elaborate set of basis strategies. With each strategy, individuals may prioritize their gains and risks according to the current situation and solve the decision-making within a smaller scope. As we live in a dynamic world that presents us with unexpected disturbances, it is also crucial to have the flexibility to alter our course of strategies accordingly. Additionally, the basis strategies can be pieced together and combined into compound strategies to reach grander goals.

For animals living in nature, the ability to flexibly formulate strategies for a complex goal is equally, if not more, crucial in their lives. Many have shown that animals exhibit complex strategy-like behaviors (*Beran et al., 2015*; *Bird and Emery, 2009*; *Brotcorne et al., 2017*; *Gruber et al., 2019*; *Leca et al., 2021*; *Loukola et al., 2017*; *Reinhold et al., 2019*; *Sabbatini et al., 2014*; *Sanz et al., 2010*),

but quantitative studies are lacking. Moreover, despite the continuing effort in studying complex behavior in animals and the underlying neural mechanisms (*Haroush and Williams, 2015*; *Kira et al., 2015*; *Ong et al., 2021*; *Yoo et al., 2020*), the level of complexity of the existing animal behavioral paradigms is insufficient for studying how animals manage strategies to simplify a sophisticated task. A sufficiently complex behavior task should allow the animal to approach an overall objective with a variety of strategies in which both the objective, its associated rewards and cost, and the behaviors can be measured and quantified. Establishing such a behavior paradigm would not only help us to understand advanced cognitive functions in animals but also lay the foundation for a thorough investigation of the underlying neural mechanism.

Here, we adapted the popular arcade game Pac-Man. The game was tweaked slightly for the macaque monkeys. Just as in the original game, the monkeys learned to use a joystick to control the movement of Pac-Man to collect all the pellets inside an enclosed maze while avoiding ghosts. The monkeys received fruit juice as a reward instead of earning points. The animals were able to learn how each element of the game led to different reward outcomes and made continuous decisions accordingly. While the game is highly dynamic and complex, it is essentially a foraging task, which may be the key to the successful training. More importantly, both the game states and the monkeys' behavior were well-defined and could be measured and recorded, providing us opportunities for quantitative analyses and modeling.

The game has a clear objective, but an optimal solution is computationally difficult. However, a set of intuitive strategies would allow players to achieve reasonable performance. To find out whether the monkeys' behavior can be decomposed into a set of strategies, we fit their gameplay with a dynamic compositional strategy model, which is inspired by recent advances in the artificial intelligence field in developing AI algorithms that solve the game with a multiagent approach (*Foderaro et al., 2017*; *Rohlfshagen et al., 2018*; *Sutton et al., 1999*; *Van Seijen et al., 2017*). The model consists of a set of simple strategies, each considering a specific aspect of the game to form decisions on how to move Pac-Man. By fitting the model to the behavior of the animals, we were able to deduce the strategy weights. The model was able to achieve over 90% accuracy for explaining the decision-making of the monkeys. More importantly, the strategy weights revealed that the monkeys adopted a take-the-best (TTB) heuristic by using a dominant strategy and only focusing on a subset of game aspects at a time. In addition, the monkeys were able to use the strategies as building blocks to form compound strategies to handle particular game situations. Our results demonstrated that animals are capable of managing a set of compositional strategies and employing hierarchical decision-making to solve a complex task.

## Results

### The Pac-Man game

We trained two monkeys to play an adapted Pac-Man (Namco) game (*Figure 1A*). In the game, the monkeys navigated a character known as Pac-Man in a maze and their objective is to traverse through the maze to eat all the pellets and energizers. The game presented the obstacles of having two ghosts named Blinky and Clyde, who behaved as predators. As in the original game, each ghost followed a unique deterministic algorithm based on Pac-Man's location and their own locations with Blinky chasing Pac-Man more aggressively. If Pac-Man was caught, the monkeys would receive a time-out penalty. Afterward, both Pac-Man and the ghosts were reset to their starting locations, and the monkeys could continue to clear the maze. If Pac-Man ate an energizer, a special kind of pellet, the ghosts would be cast into a temporary scared mode. Pac-Man could eat the scared ghosts to gain extra rewards. All the game elements that yield points in the original game provided monkeys juice rewards instead (*Figure 1A*, right). After successfully clearing the maze, the monkeys would also receive additional juice as a reward for completing a game. The fewer attempts the animals made to complete a game, the more rewards they would be given.

The game was essentially a foraging task for the monkeys. The maze required navigation, and to gain rewards, the animals had to collect pallets with the risk of encountering predators. Therefore, the game was intuitive for the monkeys, which was crucial for the training's success. The training started with simple mazes with no ghosts, and more elaborated game elements were introduced one by one

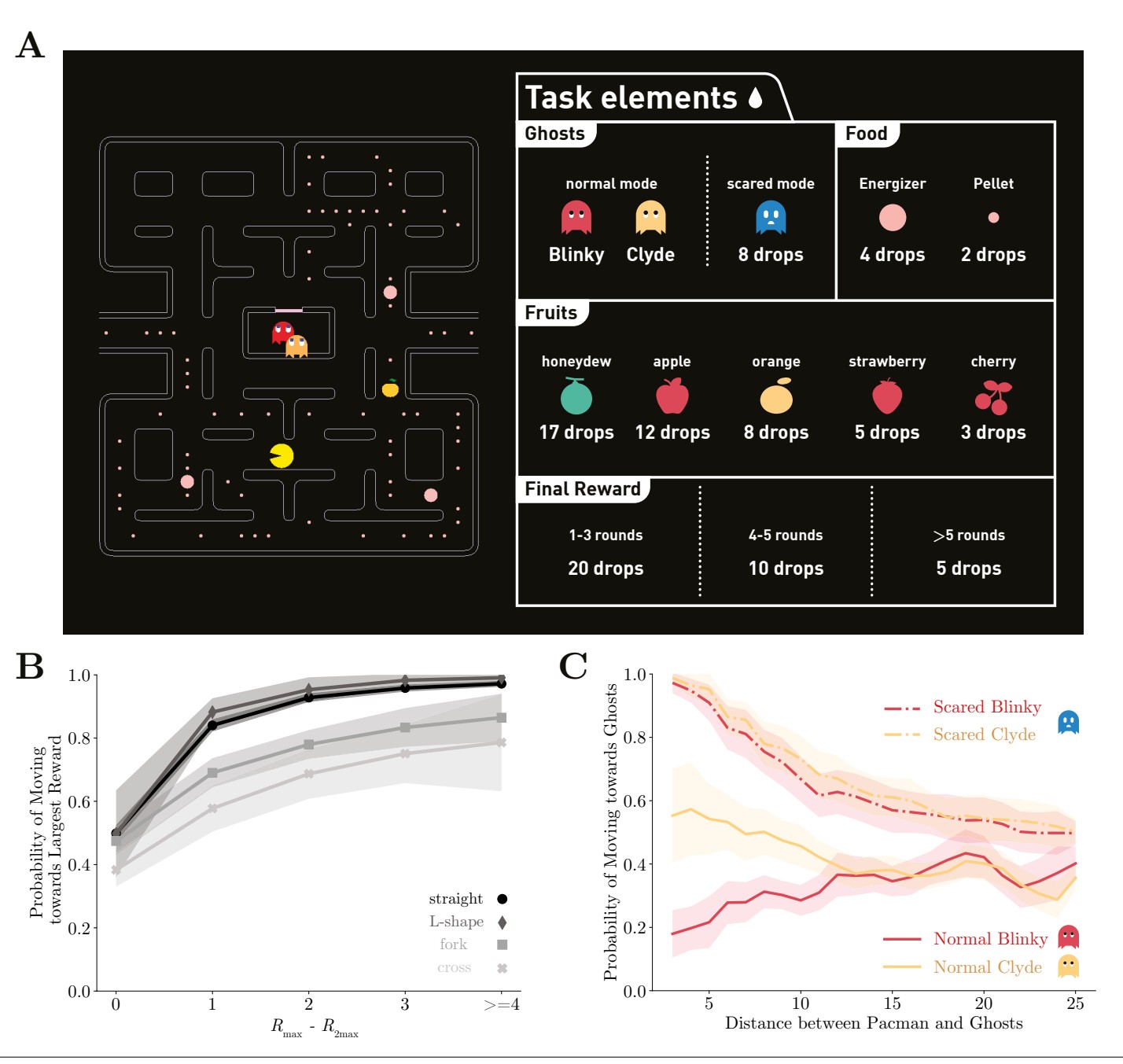

**Figure 1.** The Pac-Man game and the performance of the monkeys. (**A**) The monkeys used a joystick to navigate Pac-Man in the maze and collect pellets for juice rewards. Also in the maze, there were two ghosts, Blinky and Clyde. The maze was fixed, but the pellets and the energizers were placed randomly initially in each game. Eating energizers turned the ghosts into the scared mode for 14 s, during which they were edible. There were also fruits randomly placed in the maze. The juice rewards corresponding to each game element are shown on the right. (**B**) The monkeys were more likely to move toward the direction with more local rewards. The abscissa is the reward difference between the most and the second most rewarding direction. Different grayscale shades indicate path types with different numbers of moving directions. Means and standard errors are plotted with lines and shades. See **Figure 1—figure supplement 1** for the analysis for individual monkeys. (**C**) The monkeys escaped from normal ghosts and chased scared ghosts. The abscissa is the Dijkstra distance between Pac-Man and the ghosts. Dijkstra distance measures the distance of the shortest path between two positions on the map. Means and standard errors are denoted with lines and shades. See **Figure 1—figure supplement 1** for the analysis for individual monkeys.

The online version of this article includes the following figure supplement(s) for figure 1:

**Figure supplement 1.** The performance of Monkey O (left) and Monkey P (right).

throughout the training process (see Materials and methods and *Appendix 1—figure 1* for detailed training procedures).

The behavior analyses include data from 74 testing sessions after all the game elements were introduced to the monkeys and their performance reached a level that was subjectively determined reasonable. We recorded the joystick movements, eye movements, and pupil sizes of the animals during the game. On average, the animals completed 33 ± 9 (mean ± standard error [SE]) games in each session and each game took them 4.9 ± 1.8 attempts (*Appendix 1—figure 2*).

Optimal gameplay requires the monkeys to consider a large number of factors, and many of them vary throughout the game either according to the game rules or as a result of the monkeys' own actions. Finding the optimal strategy poses a computational challenge not only for monkeys but also for human and AI agents alike. The monkeys learned the task and played the game well, as one can see from the example games (*Appendix 1—video 1*, *Appendix 1—video 2*, *Appendix 1—video 3* for Monkey O; *Appendix 1—video 4*, *Appendix 1—video 5* for Monkey P). As a starting point to understand how the monkeys solved the task, we first studied if they understood the basic game elements, namely, the pellets and the ghosts.

First, we analyzed the monkeys' decision-making concerning the local rewards, which included the pellets, the energizers, and the fruits, within five tiles from Pac-Man for each direction. The monkeys tended to choose the direction with the largest local reward (*Figure 1B*, *Figure 1—figure supplement 1A and C*). The probability of choosing the most rewarding direction decreased with the growing number of available directions, suggesting a negative effect of option numbers on the decision-making optimality.

The monkeys also understood how to react to the ghosts in different modes. The likelihood of Pac-Man moving toward or away from the ghosts in different modes is plotted in *Figure 1C* and *Figure 1—figure supplement 1B and D*. As expected, the monkeys tended to avoid the ghosts in the normal mode and chase them when they were scared. Interestingly, the monkeys picked up the subtle difference in the ghosts' 'personalities.' By design, Blinky aggressively chases Pac-Man, but Clyde avoids Pac-Man when they get close (see Materials and methods for details). Accordingly, both monkeys were more likely to run away from Blinky but ignored or even followed Clyde when it was close by. Because the ghosts do not reverse their directions, it was actually safe for the monkeys to follow Clyde when they were near each other. On the other hand, the ghosts were treated the same by the monkeys when in scared mode. The monkeys went after the scared ghosts when they were near Pac-Man. This model-based behavior with respect to the ghosts' 'personalities' and modes suggests sophisticated decision-making of the monkeys.

These analyses suggest that the monkeys understood the basic elements of the game. While they revealed some likely strategies of the monkeys, collecting local pellets and escaping or eating the ghosts, they did not fully capture the monkeys' decision-making. Many other factors as well as the interaction between them affected the monkeys' decisions. More sophisticated behavior was required for optimal performance, and to this end, the dynamic compositional strategy model was developed to understand the monkeys' behavior.

## Basis strategies

While the overall goal of the game is to clear the maze, the monkeys may adopt different strategies for smaller objectives in different circumstances. We use the term 'strategy' to refer to the solution for these sub-goals, and each strategy involves a smaller set of game variables with easier computation for decisions that form actions.

We consider six intuitive and computationally simple strategies as the basis strategies. The *local* strategy moves Pac-Man toward the direction with the largest reward within 10 tiles. The *global* strategy moves Pac-Man toward the direction with the largest overall reward in the maze. The *energizer* strategy moves Pac-Man toward the nearest energizer. The two *evade* strategies move Pac-Man away from Blinky and Clyde in the normal mode, respectively. Finally, the *approach* strategy moves Pac-Man toward the nearest ghost. At any time during the game, monkeys could adopt one or a mixture of multiple strategies for decision-making. These basis strategies, although not necessarily orthogonal to each other (*Appendix 1—figure 3*), can be linearly combined to explain the monkeys' behavior.

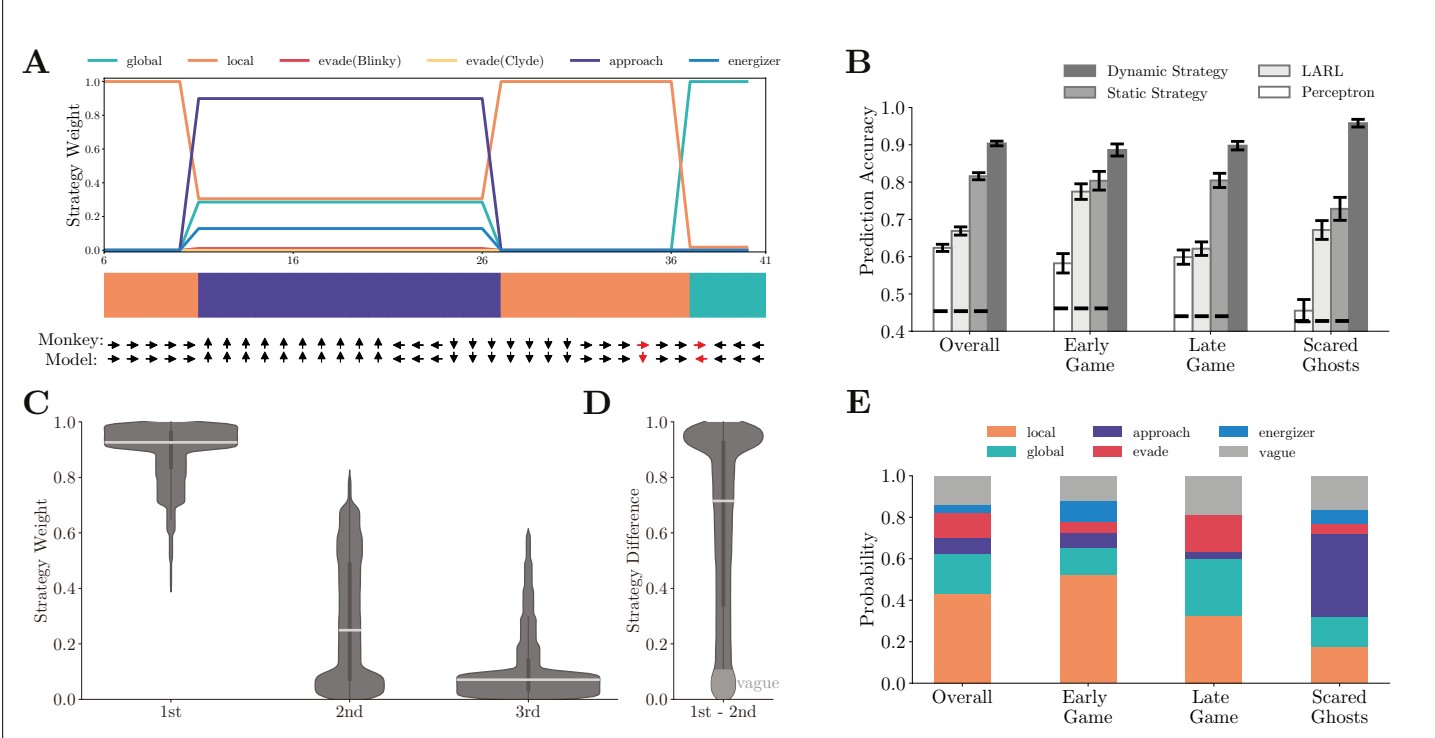

**Figure 2.** Fitting behavior with basis strategies. (**A**) The normalized strategy weights in an example game segment. The horizontal axis is the time step. Each time step is 417 ms, which is the time that it takes Pac-Man to move across a tile. The color bar indicates the dominant strategies across the segment. The monkey's actual choice and the corresponding model prediction at each time step are shown below, with red indicating a mismatch. The prediction accuracy for this segment is 0.943. Also, see *Figure 2—video 1*. (**B**) Comparison of prediction accuracy across four models in four game contexts. Four game contexts were defined according to the criteria listed in *Appendix 1—table 3*. Vertical bars denote standard deviations. Horizontal dashed lines denote the chance-level prediction accuracies. See *Appendix 1—tables 4–6* for detailed prediction accuracy comparisons. (**C**) The distribution of the three dominating strategies' weights. The most dominating strategy's weights (0.907 ± 0.117) were significantly larger than the secondary strategy (0.273 ± 0.233) and tertiary strategy (0.087 ± 0.137) by far. Horizontal white bars denote means, and the vertical black bars denote standard errors. (**D**) The distribution of the weight difference between the most and the second dominating strategies. The distribution is heavily skewed toward 1. In over 90% of the time, the weight difference was larger than 0.1, and more than 33% of the time the difference was over 0.9. (**E**) The ratios of labeled dominating strategies across four game contexts. In the early game, the *local* strategy was the dominating strategy. In comparison, in the late game, both the *local* and the *global* strategies had large weights. The weight of the *approach* strategy was largest when the ghosts were in the scared mode. See *Figure 2—figure supplement 1* for the analysis for individual monkeys.

The online version of this article includes the following video and figure supplement(s) for figure 2:

**Figure supplement 1.** Fitting behavior with strategy labels for Monkey O (left) and Monkey P (right).

**Figure 2—video 1.** Example game segment.

https://elifesciences.org/articles/74500/figures#fig2video1

At any time during the game, monkeys could adopt one or a mixture of multiple strategies for decision-making. We assumed that the final decision for Pac-Man's moving direction was based on a linear combination of the basis strategies, and the relative strategy weights were stable for a certain period. We adopted a softmax policy to linearly combine utility values under each basis strategy, with the strategy weights as model parameters. To avoid potential overfitting, we designed a two-pass fitting procedure to divide each game trial into segments and performed maximum likelihood estimation (MLE) to estimate the model parameters with the monkeys' behavior within each time segment (see Materials and methods for details). When tested with simulated data, this fitting procedure recovers the ground-truth weights used to generate the data (*Appendix 1—figure 4*).

## Monkeys adopted different strategies at different game stages

*Figure 2A* shows the normalized strategy weights in an example game segment (*Figure 2—video 1*). In this example, the monkey started with the *local* strategy and grazed pellets. With the ghosts

getting close, it initiated the *energizer* strategy and went for a nearby energizer. Once eating the energizer, the monkey switched to the *approach* strategy to hunt the scared ghosts. Afterward, the monkey resumed the *local* strategy and then used the *global* strategy to navigate toward another patch when the local rewards were depleted. The dynamic compositional strategy model (see Materials and methods for details) faithfully captures the monkey's behavior by explaining Pac-Man's movement with an accuracy of 0.943 in this example.

Overall, the dynamic compositional strategy model explains the monkeys' behavior well. The model's prediction accuracy is 0.907 ± 0.008 for Monkey O and 0.900 ± 0.009 for Monkey P. In comparison, a static strategy model, which uses the fixed strategy weights, achieves an overall accuracy of 0.806 ± 0.014 and 0.825 ± 0.012 for monkeys O and P, respectively (*Figure 2B*). The static strategy model's accuracy is still high, reflecting the fact that the monkeys were occupied with collecting pellets most of the time in the game. Thus, a combination of local and global strategies was often sufficient for explaining the monkeys' choice. However, the average accuracy measurement alone and the fixed model could not reveal the monkeys' adaptive behavior. The strategy dynamics are evident when we look at different game situations (*Figure 2B*). During the early game, defined as when there were more than 90% remaining pellets in the maze, the *local* strategy dominated all other strategies. In comparison, during the late game, defined as when there were fewer than 10% remaining pellets, both the *local* and the *global* strategies had large weights. The *approach* strategy came online when one or both scared ghosts were within 10 tiles around Pac-Man. The model's prediction accuracies for the early game, the late game, and the scared-ghosts situations were 0.886 ± 0.0016, 0.898 ± 0.011, and 0.958 ± 0.010, which were significantly higher than the static strategy model's accuracies (early: 0.804 ± 0.025, $p < 10^{-60}$; late: 0.805 ± 0.019, $p < 10^{-35}$; scared ghosts: 0.728 ± 0.031, $p < 10^{-11}$; two-sample *t*-test).

The dynamic compositional strategy decision-making model is hierarchical. A strategy is first chosen, and the primitive actions (i.e., joystick movements) are then determined under the selected strategy with a narrowed down set of features (*Botvinick et al., 2009*; *Botvinick and Weinstein, 2014*; *Dezfouli and Balleine, 2013*; *Ostlund et al., 2009*; *Sutton et al., 1999*). In contrast, in a flat model, decisions are computed directly for the primitive actions based on all relevant game features. Hierarchical models can learn and compute a sufficiently good solution much more efficiently due to its natural additive decomposition of the overall strategy utility.

To illustrate the efficiency of the hierarchical model, we tested two representative flat models. First, we considered a linear approximate reinforcement learning (LARL) model (*Sutton, 1988*; *Tsitsiklis and Van Roy, 1997*). The LARL model shared the same structure with a standard Q-learning algorithm but used the monkeys' actual joystick movements as the fitting target. To highlight the flatness of this baseline model, we adopted a common assumption that the parameterization of the utility function is linear (*Sutton and Barto, 2018*) with respect to seven game features (see Materials and methods for details). Second, we trained a perceptron network as an alternative flat model. The perceptron had one layer of 64 units for Monkey P and 16 units for Monkey O (the number of units was determined by the highest fitting accuracy with fivefold cross-validation, see Materials and methods for details). The inputs were the same features used in our six strategies, and the outputs were the joystick movements. Compared to our hierarchical models, neither flat model performed as well. The LARL model achieved 0.669 ± 0.011 overall prediction accuracy (*Figure 2B*, light gray bars) and performed worse than the hierarchical models under each game situation (early: 0.775 ± 0.021, $p < 10^{-15}$; late: 0.621 ± 0.018, $p < 10^{-17}$; scared ghosts: 0.672 ± 0.025, $p < 10^{-12}$; two-sample *t*-test). The perceptron model was even worse, both overall (0.624 ± 0.010, *Figure 2B*, white bars) and under each game situation (early: 0.582 ± 0.026, $p < 10^{-40}$; late: 0.599 ± 0.019, $p < 10^{-16}$; scared ghosts: 0.455 ± 0.030, $p < 10^{-16}$; two-sample *t*-test). The results were similar when we tested the models with individual monkeys separately (*Figure 2—figure supplement 1A and E*). Admittedly, one may design better and more complex flat models than the two tested here. Yet, even our relatively simple LARL model was more computationally complex than our hierarchical model but performed much worse, illustrating the efficiency of hierarchical models.

## Monkeys adopted TTB heuristic

Neither our model nor the fitting procedure limits the number of strategies that may simultaneously contribute to the monkeys' choices at any time, yet the fitting results show that a single strategy

often dominated the monkeys' behavior. In the example (*Figure 2A*, *Figure 2—video 1*), the monkey switched between different strategies with one dominating strategy at each time point. This was a general pattern. We ranked the strategies according to their weights at each time point. The histograms of the three dominating strategies' weights from all time points show that the most dominating strategy's weights (0.907 ± 0.117) were significantly larger than those of the secondary strategy (0.273 ± 0.233) and tertiary strategy (0.087 ± 0.137) by a significant margin (*Figure 2C*). The weight difference between the first and the second most dominating strategies was heavily skewed toward one (*Figure 2D*). Individual monkey analysis results were consistent (*Figure 2—figure supplement 1B, C, F and G*). Taken together, these results indicate that the monkeys adopted a TTB heuristics in which action decisions were formed with a single strategy heuristically and dynamically chosen.

Therefore, we labeled the monkeys' strategy at each time point with the dominating strategy. When the weight difference between the dominating and secondary strategies was smaller than 0.1, the strategy was labeled as *vague*. It may reflect a failure of the model to identify the correct strategy, a period of strategy transition during which a dominating strategy is being formed to replace the existing one, or a period during which the monkeys were indeed using multiple strategies. No matter which is the case, they are only a small percentage of data and not representative.

The *local* and the *global* strategy were most frequently used overall. The *local* strategy was particularly prevalent during the early game when the local pellets were abundant, while the *global* strategy contributed significantly during the late game when the local pellets were scarce (*Figure 2E*). Similar strategy dynamics were observed in the two monkeys (*Figure 2—figure supplement 1D and H*).

## Strategy manifested in behavior

The strategy fitting procedure is indifferent to how the monkeys chose between the strategies, but the fitting results provide us with some hints. The probability of the monkeys adopting the *local* or the *global* strategy correlated with the availability of local rewards: abundant local rewards lead to the *local* strategy (*Figure 3A*, individual monkeys: *Figure 3—figure supplement 1A and E*). On the other hand, when the ghosts were scared, the decision between chasing the ghosts and going on collecting the pellets depended on the distance between Pac-Man and the scared ghosts (*Figure 3B*, individual monkeys: *Figure 3—figure supplement 1B and F*). In addition, during the *global* strategy, the monkeys often moved Pac-Man to reach a patch of pellets far away from its current location. They chose the shortest path (*Figure 3C*, individual monkeys: *Figure 3—figure supplement 1C and G*) and made the fewest turns to do so (*Figure 3D*, individual monkeys: *Figure 3—figure supplement 1D and H*), demonstrating their goal-directed path-planning behavior under the particular strategy.

The fitting results can be further corroborated from the monkeys' eye movements and pupil dilation. Because different game aspects were used in different strategies, the monkeys should be looking at different things when using different strategies. We classified monkeys' fixation locations into four categories: ghosts, energizers, pellets, and others (see Materials and methods for details). *Figure 3E* (individual monkeys: *Figure 3—figure supplement 2A–E*) shows the fixation ratio of these game objects under different strategies. Although a large number of fixations were directed at the pellets in all situations, they were particularly frequent under the *local* and *energizer* strategies. Fixations directed to the energizers were scarce, unless when the monkeys adopted the *energizer* strategy. On the other hand, monkeys looked at the ghosts most often when the monkeys were employing the *approach* strategy to chase the ghosts (p<0.001, two-sample *t*-test). Interestingly, the monkeys also looked at the ghosts more often under the *energizer* strategy than under the *local* strategy (p<0.001, two-sample *t*-test), which suggests that the monkeys were also keeping track of the ghosts when going for the energizer.

While the fixation patterns revealed that the monkeys paid attention to different game elements in different strategies, we also identified a physiological marker that reflected the strategy switches in general but was not associated with any particular strategy. Previous studies revealed that non-luminance-mediated changes in pupil diameter can be used as markers of arousal, surprise, value, and other factors during decision-making (*Joshi and Gold, 2020*). Here, we analyzed the monkeys' pupil dilation during strategy transitions. When averaged across all types of transitions, the pupil diameter exhibited a significant but transient increase around strategy transitions (p<0.01, two-sample *t*-test, *Figure 3F*, also individual monkeys: *Figure 3—figure supplement 2B–F*). Such an increase was absent when the strategy transition went through a *vague* period. This increase was evident in the

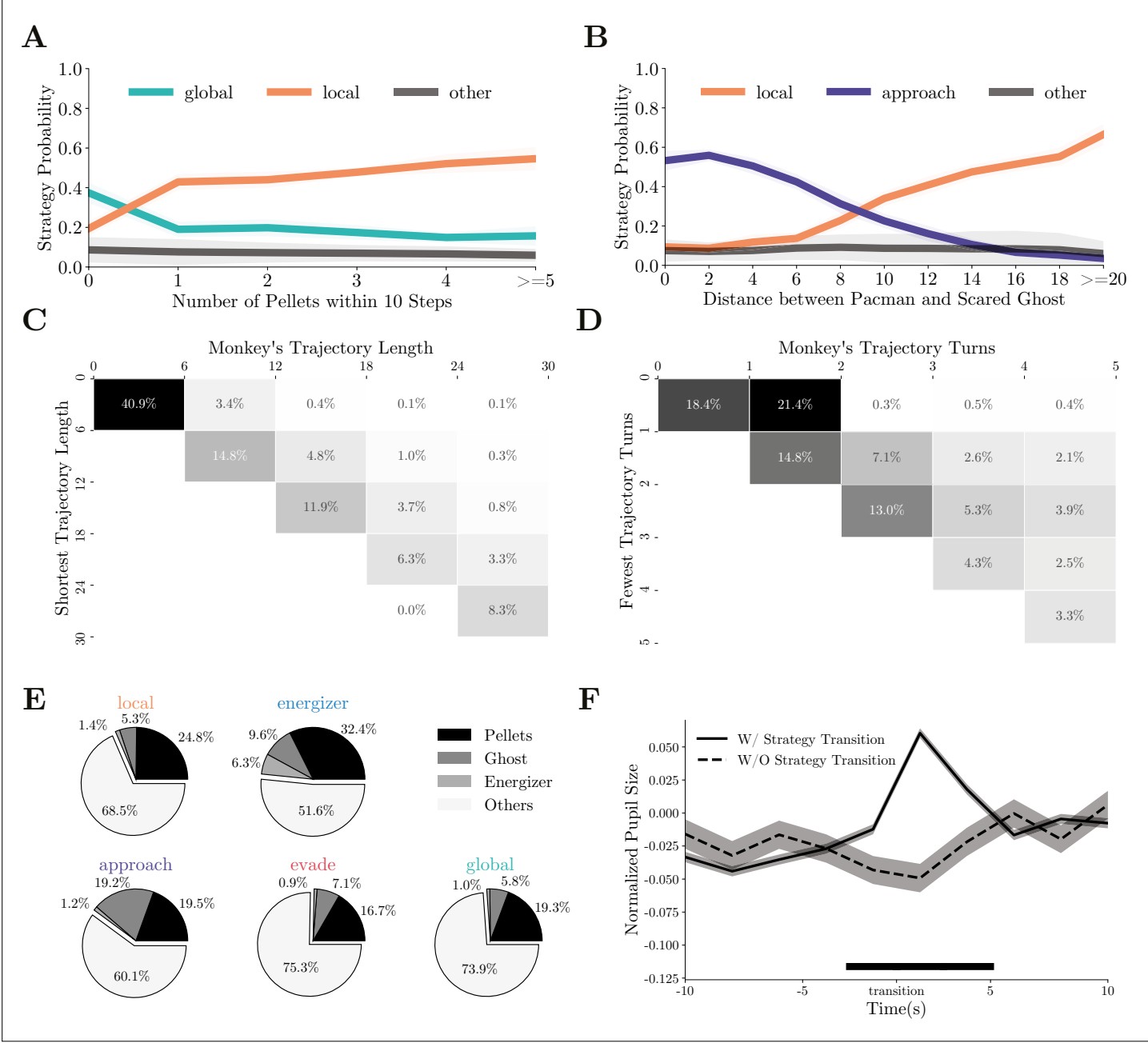

**Figure 3.** Monkeys' behavior under different strategies. (**A**) The probabilities of the monkeys adopting the *local* or *global* strategy correlate with the number of local pellets. Solid lines denote means, and shades denote standard errors. (**B**) The probabilities of the monkeys adopting the *local* or *approach* strategy correlate with the distance between Pac-Man and the ghosts. Solid lines denote means, and shades denote standard errors. (**C**) When adopting the *global* strategy to reach a far-away patch of pellets, the monkeys' actual trajectory length was close to the shortest. The column denotes the actual length, and the row denotes the optimal number. The percentages of the cases with the corresponding actual lengths are presented in each cell. High percentages in the diagonal cells indicate close to optimal behavior. (**D**) When adopting the *global* strategy to reach a far-away patch of pellets, the monkeys' number of turns was close to the fewest possible turns. The column denotes the actual turns, and the row denotes the optimal number. The percentages of the cases with the corresponding optimal numbers are presented in each cell. High percentages in the diagonal cells indicate close to optimal behavior. (**E**) Average fixation ratios of ghosts, energizers, and pellets when the monkeys used different strategies. (**F**) The monkeys' pupil diameter increases around the strategy transition (solid line). Such increase was absent if the strategy transition went through the *vague* strategy (dashed line). Shades denote standard errors. Black bar at the bottom denotes p<0.01, two-sample *t*-test. See *Figure 3—figure supplement 1* for the analysis for individual monkeys.

The online version of this article includes the following figure supplement(s) for figure 3:

*Figure 3 continued on next page*

*Figure 3 continued*

**Figure supplement 1.** Monkey's behavior under different strategies for Monkey O (left) and Monkey P (right).

**Figure supplement 2.** Monkey's eye movement patterns under different strategies for Monkey O (left) and Monkey P (right).

transitions in both directions, for example, from *local* to *global* (*Figure 3—figure supplement 2C–G*) and from *global* to *local* (*Figure 3—figure supplement 2D–H*). Therefore, it cannot be explained by any particular changes in the game state, such as the number of local pellets. Instead, it reflected a computation state of the brain associated with strategy switches (*Nassar et al., 2012*; *Urai et al., 2017*; *Wang et al., 2021*).

## Compound strategies

The compositional strategy model divides the monkeys' decision-making into different hierarchies (*Figure 4*). At the lowest level, decisions are made for actions, the joystick movements of up, down, left, and right, using one of the basis strategies. At the middle level, decisions are made between these basis strategies, most likely with a heuristic for the monkeys to reduce the complexity of the decision-making. The pupil dilation change reflected the decision-making at this level. In certain situations, the basis strategies may be pieced together and form compound strategies as a higher level of decision-making. These compound strategies are not simple impromptu strategy assemblies. Instead, they may reflect more advanced planning. Here, building on the strategy analyses, we describe two scenarios in which compound strategies were used by the monkeys.

The first scenario involves the energizers, which is an interesting feature of the game. They not only provide an immediate reward but also lead to potential future rewards from eating ghosts. With the knowledge that the effect of the energizers was only transient, the monkeys could plan accordingly to maximize their gain from the energizers. In some trials, the monkeys immediately switched to the *approach* strategy after eating an energizer and actively hunted nearby ghosts (*Figure 5—video 1*). In contrast, sometimes the monkey appeared to treat an energizer just as a

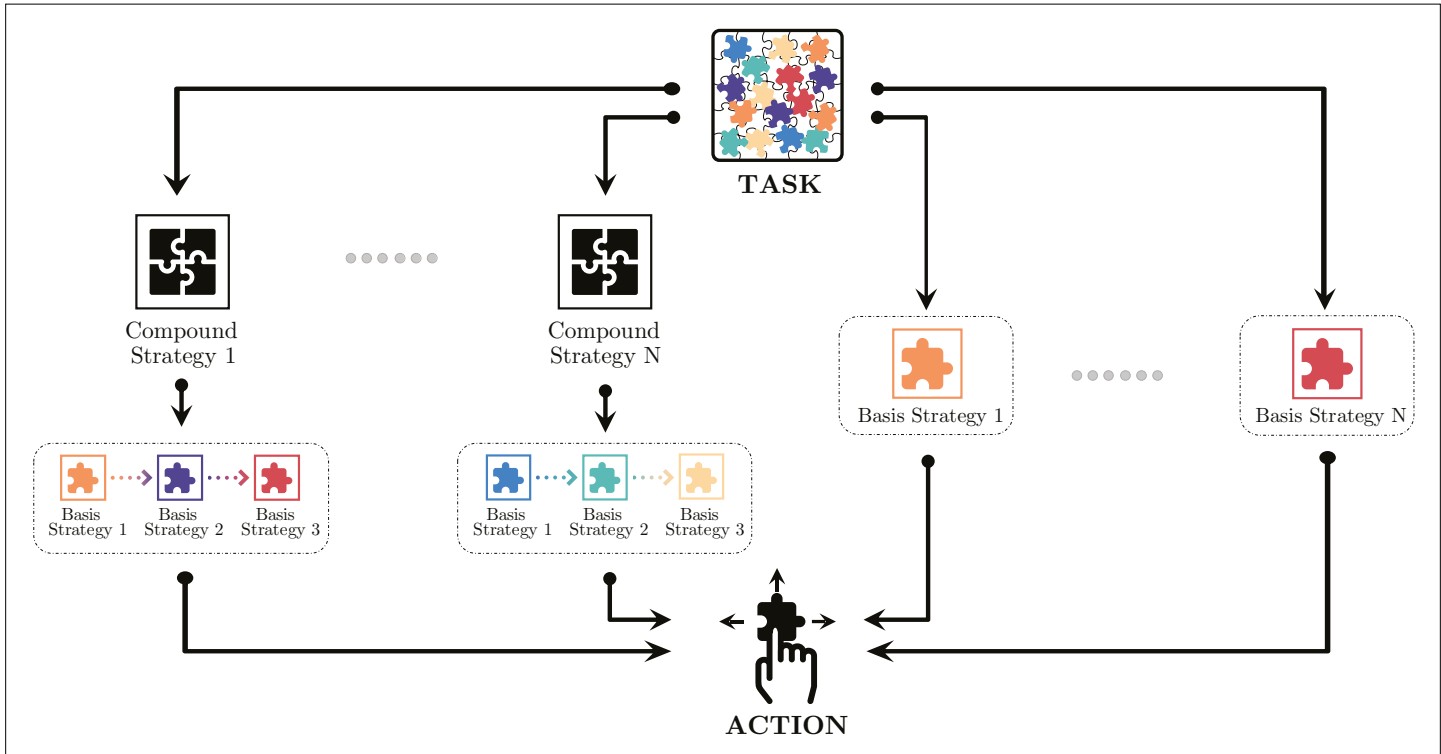

**Figure 4.** Monkeys' decision-making in different hierarchies. At the lowest level, decisions are made for the joystick movements: up, down, left, or right. At the middle level, choices are made between the basis strategies. At a higher level, simple strategies may be pieced together for more sophisticated compound strategies. Monkeys may adopt one of the compound strategies or just a basis strategy depending on the game situation.

more rewarding pellet. They continued collecting pellets with the *local* strategy after eating the energizer, and catching a ghost seemed to be accidental and unplanned (***Figure 5—video 2***). Accordingly, we distinguished these two behaviors using the strategy labels after the energizer consumption and named the former as *planned attack* and the latter as *accidental consumption* (see Materials and methods for details). With this criterion, we extracted 493 (Monkey O) and 463 (Monkey P) *planned attack* plays, and 1970 (Monkey O) and 1295 (Monkey P) *accidental consumption* plays in our dataset.

The strategy weight dynamics around the energizer consumption showed distinct patterns when the animals adopted the compound strategy *planned attack* (***Figure 5A***, individual monkeys: ***Figure 5—figure supplement 1A–E***). When the monkeys carried out *planned attacks*, they started to approach the ghosts well before the energizer consumption, which is revealed by the larger weights of the *approach* strategy than that in the *accidental consumption*. The monkeys also cared less for the local pellets in *planned attacks* before the energizer consumption. The weight dynamics suggest that the decision of switching to the *approach* strategy was not an afterthought but planned well ahead. The monkeys strung the *energizer/local* strategy with the *approach* strategy together into the compound strategy to eat an energizer and then hunt the ghosts. Such a compound strategy should only be employed when Pac-Man, ghosts, and an energizer are in close range. Indeed, the average of distance between Pac-Man, the energizer, and the ghosts was significantly smaller in *planned attack* than in *accidental consumption* (***Figure 5B***, p<0.001, two-sample *t*-test, individual monkeys: ***Figure 5—figure supplement 1B–F***).

Again, the *planned attacks* were also associated with distinct eye movement and pupil size dynamics. The monkeys fixated on the ghosts, the energizers, and Pac-Man more frequently before the energizer consumption in *planned attacks* than in *accidental consumption* (***Figure 5C***, p<0.001, two-sample *t*-test, individual monkeys: ***Figure 5—figure supplement 1C–G***), reflecting more active planning under the way. In addition, the monkeys' pupil sizes were smaller before they caught a ghost in *planned attacks* than in *accidental consumption* (p<0.01, two-sample *t*-test), which may reflect a lack of surprise under *planned attacks* (***Figure 5D***, individual monkeys: ***Figure 5—figure supplement 1D–H***). The difference was absent after the ghost was caught.

The second scenario involves a counterintuitive move in which the monkeys moved Pac-Man toward a normal ghost to die on purpose in some situations. Although the move appeared to be suboptimal, it was beneficial in a certain context. The death of Pac-Man resets the game and returns Pac-Man and the ghosts to their starting positions in the maze. As the only punishment in the monkey version of the game is a time-out, it is advantageous to reset the game by committing such suicide when local pellets are scarce, and the remaining pellets are far away.

To analyze this behavior, we defined the compound strategy *suicide* using strategy labels. We computed the distances between Pac-Man and the closest pellets before and after its death. In *suicides*, Pac-Man's death significantly reduced this distance (***Figure 6A***, upper histogram, ***Figure 6—video 1***, individual monkeys: ***Figure 6—figure supplement 1A–E***). This was not true when the monkeys were adopting the *evade* strategy but failed to escape from the ghosts (*failed evasions*, ***Figure 6A***, bottom histogram, ***Figure 6—video 2***, individual monkeys: ***Figure 6—figure supplement 1A–E***). In addition, the distance between Pac-Man and the ghosts was greater in *suicides* (***Figure 6B***, p<0.001, two-sample *t*-test, individual monkeys: ***Figure 6—figure supplement 1B and F***). Therefore, these suicides were a proactive decision. Consistent with the idea, the monkeys tended to saccade toward the ghosts and pellets more often in *suicides* than in *failed evasions* (***Figure 6C***, p<0.001, two-sample *t*-test, individual monkeys: ***Figure 6—figure supplement 1C and G***). Their pupil size decreased even before Pac-Man's death in *suicides*, which was significantly smaller than in *failed evasions*, suggesting that the death was anticipated (***Figure 6D***, p<0.01, two-sample *t*-test, individual monkeys: ***Figure 6—figure supplement 1D and H***).

Together, these two examples demonstrated how monkeys' advanced gameplay can be understood with concatenated basis strategies. The compositional strategy model not only provides a good fit for the monkeys' behavior but also offers insights into the monkeys' gameplay. The compound strategies demonstrate that the monkeys learned to actively change the game into desirable states that can be solved with planned strategies. Such intelligent behavior cannot be explained with a passive foraging strategy.

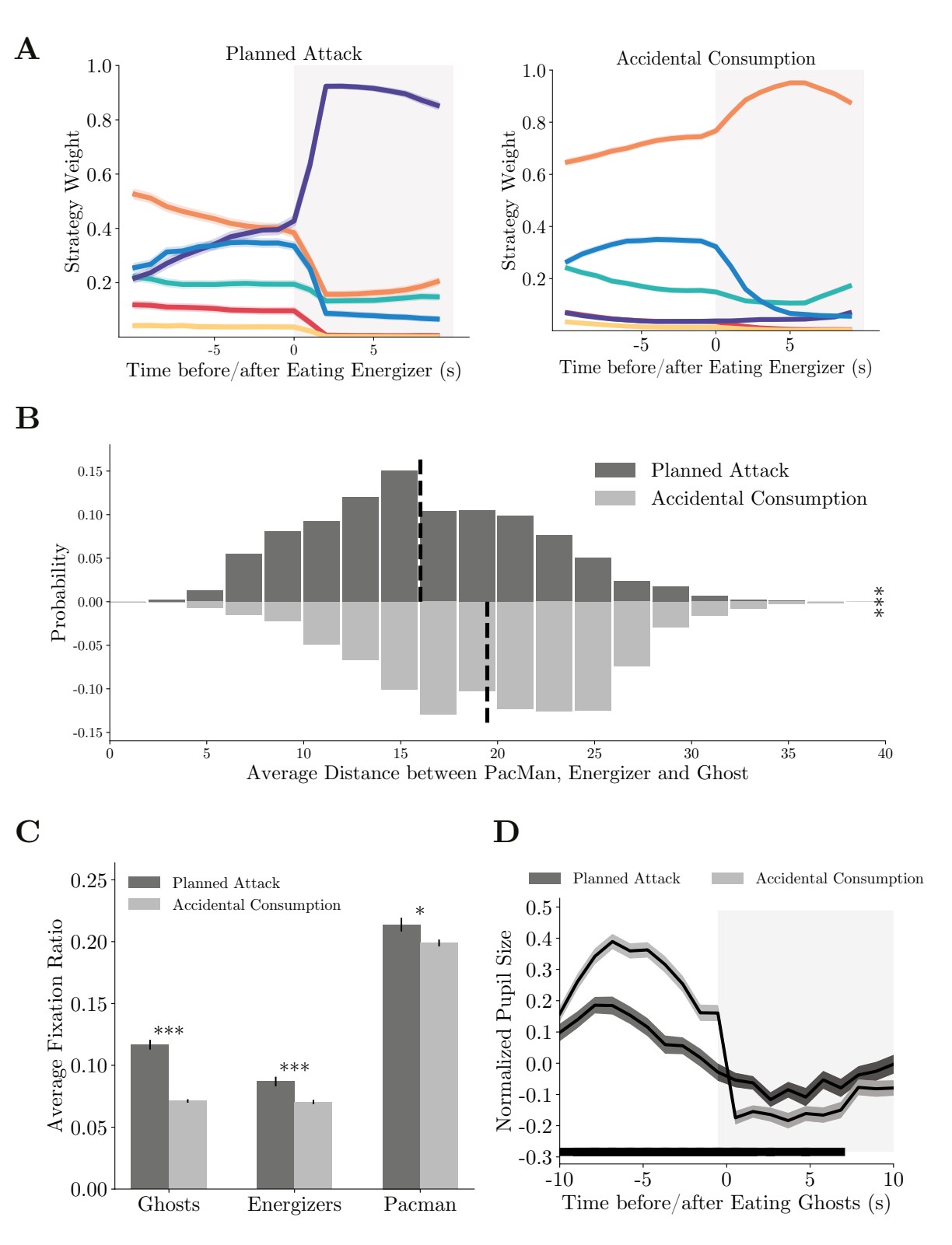

**Figure 5.** Compound strategies: *planned attack*. (**A**) Average strategy weight dynamics in *planned attacks* (left) and *accidental consumptions* (right). Solid lines denote means, and shades denote standard errors. (**B**) The average distance between Pac-Man, the energizer, and the ghosts in *planned attacks* and *accidental consumptions*. Vertical dashed lines denote means. ***p<0.001, two-sample *t*-test. (**C**) Ratios of fixations on the ghosts, the energizer, and Pac-Man. Vertical bars denote standard errors. ***p<0.001, **p<0.01, two-sample *t*-test. (**D**) The pupil size aligned to the ghost

*Figure 5 continued on next page*

*Figure 5 continued*

consumption. The black bar near the abscissa denotes data points where the two traces are significantly different (p<0.01, two-sample *t*-test). Shades denote standard errors at every time point. See *Figure 5—figure supplement 1* for the analysis for individual monkeys.

The online version of this article includes the following video and figure supplement(s) for figure 5:

**Figure supplement 1.** *Planned attacks* in Monkey O (upper) and Monkey P (lower).

**Figure 5—video 1.** *Planned attack* game segment.

https://elifesciences.org/articles/74500/figures#fig5video1

**Figure 5—video 2.** *Accidental consumption* game segment.

https://elifesciences.org/articles/74500/figures#fig5video2

## Discussion

Just as one cannot gain a full understanding of the visual system by studying it with bars and dots, pursuing a deeper insight into the cognitive capability of the brain demands sophisticated behavior paradigms in which an ensemble of perception, attention, valuation, executive control, decision-making, motor planning, and other cognitive processes need to work together continuously across time. Naturally, quantifying and modeling these behaviors in such paradigms is challenging, but here we demonstrated that the behavior of monkeys during a complex game can be understood and described with a set of basis strategies that decompose the decision-making into different hierarchies.

Our hierarchical model explains the monkeys' joystick movements well (*Figure 2B*). Importantly, the strategies derived from the model can be verified with independent behavior measurements. The monkeys' fixation pattern, a measure of their attention, reflected the features associated with the current strategy (*Figure 3E*). Moreover, an increase in pupil dilation (*Figure 3F*), which was not associated with any particular changes of game states, was found at the deduced strategy switches. This is consistent with the prediction from the hierarchical model that there should be crucial decision-making of strategies around the strategy transitions.

In contrast to hierarchical models in which the decision-maker divides decision-making into multiple levels and at each level focuses on an increasingly refined smaller set of game features (*Botvinick et al., 2009*; *Botvinick and Weinstein, 2014*; *Dezfouli and Balleine, 2013*; *Ostlund et al., 2009*; *Sutton et al., 1999*), a flat model's decisions are directly computed at the primitive action level, and each action choice is evaluated with all game features. Although in theory a flat model may achieve equal or even greater performance than a hierarchical model, flat models are much more computationally costly. Especially when working memory has a limited capacity, as in the case of the real brain, hierarchical models can achieve a faster and more accurate performance (*Botvinick and Weinstein, 2014*). Our Pac-Man task contains an extensive feature space while requiring real-time decision-making that composes limitations on the cognitive resources. Even for a complex flat model such as Deep Q-Network, which evaluates primitive actions directly with a deep learning network structure without any temporally extended higher-level decisions (*Mnih et al., 2015*), the game performance is much worse than a hierarchical model (*Van Seijen et al., 2017*). In fact, the most successful AI player to date uses a multiagent solution, which is hierarchical in nature (*Van Seijen et al., 2017*). Our study shows that the monkeys also adopted a hierarchical solution for the Pac-Man game.

Although the particular set of basis strategies in the model is hand-crafted, we have good reasons to believe that they reflect the decision-making of the monkeys. Our model fitting procedure is agnostic to how one should choose between the strategies, yet the resulting strategies can be corroborated both from monkeys' route planning, eye movements, and pupil dilation patterns. This is evidence for both the validity of the model and the rationality behind monkeys' behavior. The correlation between the results of strategy fitting and the fixation patterns of the monkeys indicates that the animals learned to selectively attend to the features that were relevant for their current strategy while ignoring others to reduce the cognitive load for different states. Similar behaviors have also been observed in human studies (*Leong et al., 2017*; *Wilson and Niv, 2011*). In particular, the pupil dilation at the time of strategy transitions indicated the extra cognitive processing carried out in the brain to handle the strategy transitions. Lastly, the model, without being specified so, revealed that a single strategy dominates monkeys' decision-making during most of the game. This is consistent with the idea that strategy-using is a method that the brain uses to simplify decision-making by ignoring irrelevant game aspects to solve complex tasks (*Binz et al., 2022*; *Moreno-Bote et al., 2020*).

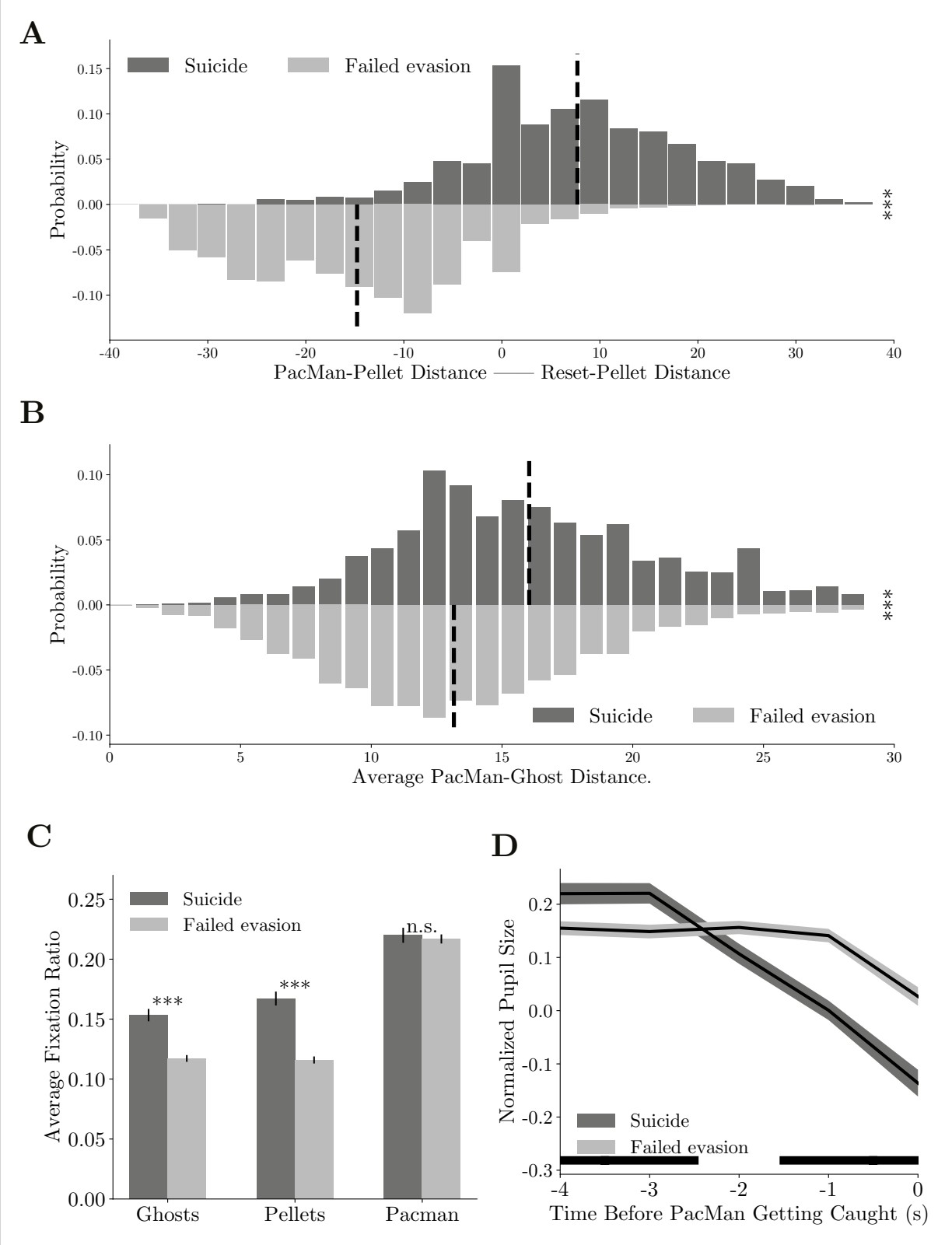

**Figure 6.** Compound strategies: *suicide*. (**A**) Distance difference between Pac-Man and closest pellet before and after the death is smaller in *suicides* than in *failed evasions*. Vertical dashed lines denote means. \*\*\*p<0.001, two-sample *t*-test. (**B**) Average distance between Pac-Man and the ghosts was greater in *suicides* than in *failed evasions*. Vertical dashed lines denote means. \*\*\*p<0.001, two-sample *t*-test. (**C**) The monkeys fixated more frequently on the ghosts and the pellets in *suicides* than in *failed evasions*. Vertical bars denote standard errors. \*\*\*p<0.001, two-sample *t*-test. (**D**) The monkeys'

*Figure 6 continued on next page*

*Figure 6 continued*

pupil size decreased before Pac-Man's death in *suicides*. The black bar near the abscissa denotes data points where the two traces are significantly different (p<0.01, two-sample *t*-test). Shades denote standard errors at every time point. See *Figure 6—figure supplement 1* for the analysis for individual monkeys.

The online version of this article includes the following video and figure supplement(s) for figure 6:

**Figure supplement 1.** *Suicides* in Monkey O (left) and Monkey P (right).

**Figure 6—video 1.** *Suicide* game segment.

https://elifesciences.org/articles/74500/figures#fig6video1

**Figure 6—video 2.** *Failed evasion* game segment.

https://elifesciences.org/articles/74500/figures#fig6video2

In some previous animal studies, strategies were equated to decision rules (*Bunge and Wallis, 2007*; *Genovesio and Wise, 2007*; *Hoshi et al., 2000*; *Mante et al., 2013*; *Tsujimoto et al., 2011*). The rules were typically mutually exclusive, and the appropriate rule was either specified with explicit sensory cues or the trial-block structure. The rules in these studies can be boiled down to simple associations, even in cases when the association may be abstract (*Genovesio and Wise, 2007*). In this study, however, we defined a set of strategies as a heuristic that reduces a complex computation into a set of smaller and more manageable problems or computations. There were no explicit cues or trial-block structures to instruct animals on which strategies to choose. Nevertheless, the same prefrontal network, including the dorsolateral prefrontal cortex, orbitofrontal cortex, and polar cortex that are suggested to engage in rule use and rule switching (*Bunge and Wallis, 2007*; *Genovesio and Wise, 2007*; *Hoshi et al., 2000*; *Mante et al., 2013*; *Tsujimoto et al., 2011*), may also play important roles in strategy-based decision-making, too.

Our Pac-Man paradigm elicits monkeys' more complex and natural cognitive ability. First, the game contains an extensive state space. This requires monkeys to simplify the task by developing temporally extended strategies to accomplish sub-goals. Second, there exists nonexclusive solutions or strategies to solve the Pac-Man task appropriately. Instead of spoon-feeding monkeys the exact solution in simple tasks, we trained them with all relevant game elements during the training phases and allowed them to proactively coordinate and select strategies freely. Therefore, our Pac-Man paradigm does not restrict monkeys' behavior with a small number of particular rules and allows the brain and its neural circuitry to be studied in a more natural setting (*Krakauer et al., 2017*).

In summary, our model distilled a complex task into different levels of decision-making centered around a set of compositional strategies, which paved the way for future experiments that will provide key insights into the neural mechanisms underlying sophisticated cognitive behavior that go beyond what most of the field currently studies.

## Materials and methods

### Subjects and materials

Two male rhesus monkeys (*Macaca mulatta*) were used in the study (O and P). They weighed on average 6–7 kg during the experiments. All procedures followed the protocol approved by the Animal Care Committee of Shanghai Institutes for Biological Sciences, Chinese Academy of Sciences (CEBSIT-2021004).

### Training procedure

To help monkeys understand the Pac-Man game and develop their decision-making strategies, we divided the training procedures into the following three stages. In each stage, we gradually increased game depth based on their conceptual and implementational complexity.

### Stage 1: Reward

In the first stage, the monkeys were trained to use the joystick to control Pac-Man to navigate in simple mazes for collecting pellets (*Appendix 1—figure 1A*). Training began with the horizontal and the vertical linear mazes. In each maze, Pac-Man started from the center where pellets were at one end and a static ghost was at the other end. Monkeys earned two drops of juice (one drop = 0.5 mL)

immediately when consuming a pellet. Monkeys could earn an extra-large amount of juice by clearing all pellets. Running toward the static ghost would lead to the end of the trial with a time-out penalty (5 s). When the monkeys completed more than 100 correct trials with above 80% accuracy, we introduced two slightly more complex mazes, the T and the upside-down T maze. After the monkeys completed more than 50 correct trials in the T-mazes with above 80% accuracy, we introduced the H-maze. Stage 1 training included 58 sessions for Monkey O and 84 sessions for Monkey P.

### Stage 2: Ghost
In the second stage, the monkeys were trained to deal with the ghosts (*Appendix 1—figure 1B*). In addition, the mazes used in this stage were closed and had loops. A ghost would block one of the routes leading to the pellets, forcing the monkeys to take alternative routes. In the first phase, the ghost was stationary in a square maze. Pac-Man started from one of the four corners, and pellets were distributed in the two adjacent arms. The ghost was placed at the corner where the two arms joined, forcing Pac-Man to retreat after clearing the pellets in one arm. In the second phase, the ghost moved within the arm. In the third phase, the ghost would chase Pac-Man. Stage 2 training included 86 sessions for Monkey O and 74 sessions for Monkey P.

### Stage 3: Energizer
In this stage, the monkeys were trained to understand the energizer (*Appendix 1—figure 1C*). In the first phase, the monkeys were trained to understand the distinction between normal and scared ghosts. We used the square maze with a normal or a scared ghost randomly placed across trials. Blinky in the normal mode would chase Pac-Man, while in scared mode move in random directions at half of Pac-Man's speed. Monkeys earned eight drops of juice after eating a scared ghost. In the second phase, the monkeys were trained with the maze that the scared mode could only be triggered by an energizer. Two energizers were randomly placed in each maze. Monkeys earned four drops of juice when eating an energizer and turned ghosts into the scared mode immediately. The scared mode lasted 14 s. As a reminder, ghosts in the scared mode flashed for 2 s before turning back into the normal mode. In the third phase, we adopted the maze used in our final gameplay recording (*Figure 1A*). The detailed game rules can be found in the following 'Task paradigm' session. Stage 3 training included 248 sessions for Monkey O and 254 sessions for Monkey P.

## Task paradigm
The Pac-Man game in this study was adapted from the original game by Namco. All key concepts of the game are included. In the game, the monkey navigates a character named Pac-Man through a maze with a four-way joystick to collect pellets and energizers. The maze is sized at 700 × 900 pixels, displayed at the resolution of 1920 × 1080 on a 27-inch monitor placed at 68 cm away from the monkey. The maze can be divided into square tiles of 25 × 25 pixel.(*Binz et al., 2022*). The pellets and energizers are placed at the center of a tile, and they are consumed when Pac-Man moves into the tile. In the recording sessions, there are 88 or 73 pellets in the maze, each worth two drops of juice, and three or four energizers, each worth four drops of juice. We divided the maze into four quarters, and the pellets and energizers are randomly placed in three of them, with one randomly chosen quarter empty. In addition, just as in the original game, there are five different kinds of fruits: cherry, strawberry, orange, apple, and melon. They yield 3, 5, 8, 12, and 17 drops of juice, respectively. In each game, one randomly chosen fruit is placed at a random location at each game. As in the original game, the maze also contains two tunnels that teleport Pac-Man to the opposite side of the maze.

There are two ghosts in the game, Blinky and Clyde. They are released from the Ghost Home, which is the center box of the maze, at the beginning of each game. Blinky is red and is more aggressive. It chases Pac-Man all the time. Clyde is orange. It moves toward Pac-Man when it is more than eight tiles away from Pac-Man. Otherwise, it moves toward the lower-left corner of the maze. The eyes of the ghosts indicate the direction they are traveling. The ghosts cannot abruptly reverse their direction in the normal mode. Scared ghosts move slowly to the ghost pen located at the center of the maze. The scared state lasts 14 s, and the ghosts flash as a warning during the last 2 s of the scared mode. Monkeys get eight drops of juice if they eat a ghost. Dead ghosts move back to the ghost pen and then respawn. The ghosts can also move through the tunnels, but their speed is reduced when

inside the tunnel. For more explanations on the ghost behavior, please refer to https://gameinternals.com/understanding-pac-man-ghost-behavior.

When Pac-Man is caught by a ghost, it and the ghosts return to the starting location. The game is restarted after a time-out penalty. When all the pellets and energizers are collected, the monkey receives a reward based on the number of rounds that takes the monkey to complete the game: 20 drops if the round number is from 1 to 3; 10 drops if the round number is from 4 to 5; 5 drops if the round number is larger than 5.

## Behavioral data recording and preprocessing

We monitored the monkeys' joystick movements, eye positions, and pupil sizes during the game. The joystick movements were sampled at 60 Hz. We used Eyelink 1000 Plus to record two monkeys' eye positions and pupil sizes. The sampling rate was 500 Hz or 1000 Hz.

The data we presented here are based on the sessions after the monkeys went through all the training stages and were able to play the game consistently. On average, monkeys completed 33 ± 9 games in each session and each game took them 4.86 ± 1.75 attempts. The dataset includes 3217 games, 15,772 rounds, and 899,381 joystick movements. The monkeys' detailed game statistics are shown in *Appendix 1—figure 2*.

## Basic performance analysis

In *Figure 1B*, we compute the rewards for each available moving direction at each location by summing up the rewards from the pellets (one unit) and the energizers (two units) within five steps from Pac-Man's location. Locations are categorized into four path types defined in *Appendix 1—table 1*. For each path type, we calculate the probability that the monkey moved in the direction with the largest rewards conditioned on the reward difference between the most ($R_{max}$) and the second most rewarding directions ($R_{2max}$). In *Figure 1C*, we compute the likelihood of Pac-Man moving toward or away from the ghosts in different modes with different Dijkstra distance between Pac-Man and the ghosts. We classified Pac-Man's moving action into two types, toward and away, according to whether the action decreased or increased the Dijkstra distance between Pac-Man and the ghosts. Dijkstra distance is defined as the distance of the shortest path between two positions in the maze.

## Basis strategies

We include six basis strategies in the hierarchical strategy model. In each basis strategy, we compute the utility values for all directions ($\mathcal{D}$={left, right, up, down}), expressed as a vector of length 4. Notice that not all directions are available, utility values for unavailable directions are set to be negative infinity. The moving direction is computed according to the largest average utility value for each strategy.

We determine the utility associated with each direction and its possible trajectories. Specifically, let $p$ represents Pac-Man's position and $\tau(p)$ represent a path starting from $p$ with the length of 10. We define $g = g_B, g_C$ to be the position of two ghosts, Blinky and Clyde, and $r = r_p, r_e, r_f$ to be the positions of pellets, energizers, and fruits, respectively. We compute the utility of each path $\tau(p)$ as follows (without specific noting, $\tau(p)$ is denoted as $\tau$ for simplicity).

We use the *local* strategy to describe the local graze behavior within a short distance with the utility function defined as

$$U(\tau) = \sum_{r' \in \tau \cap r_e, r_p, r_f} \text{Reward}(r') \tag{1}$$

where $\tau \cap r_e, r_p, r_f$ denotes the pellets/energizers/fruits on the path. Specific parameters for awarded and penalized utilities of each game element in the model can be found in *Appendix 1—table 2*.

*Evade* strategy focuses on dodging close-by ghosts. Specifically, we create two evade strategies (*evade* Blinky and *evade* Clyde) that react to the respective ghost, with the utility function defined as

$$U(\tau) = \begin{cases} \Im(g \text{ is normal}) \times \text{Penalty}(g), & \text{if } g \in \tau \\ 0, & \text{otherwise} \end{cases} \tag{2}$$

with $g = g_B$ and $g = g_C$, respectively. Here, $\Im(s)$ is an indication function, where $\Im(s) = 1$ when statement $s$ is true, otherwise $\Im(s) = 0$.

*Energizer* strategy moves Pac-Man toward the closest energizer. In this case, the rewards set $r$ only contains the positions of energizers (i.e., $r = r_e$):

$$U(\tau) = \sum_{r' \in \tau \cap r_e} \text{Reward}(r') \tag{3}$$

*Approach* strategy moves Pac-Man toward the ghosts, regardless of the ghosts' mode. Its utility function is

$$U(\tau) = \sum_{r' \in \tau \cap g_B, g_C} \text{Reward}(r') \tag{4}$$

*Global* strategy does not use a decision tree. It counts the total number of pellets in the whole maze in each direction without considering any trajectories. For example, the utility for the down direction is the total number of pellets that sit vertically below Pac-Man's location.

We construct the utility of each agent as a vector $U_a \in R^4$, for $a \in \{local, global, evade \text{ Blinky}, evade \text{ Clyde}, approach, energizer\}$ of the four directions. For each direction $d \in \mathfrak{D}$, its utility is obtained by averaging utilities $U(\tau)$ on all the path sets $\tau \in \mathcal{T}$ in that direction:

$$\mathrm{U}_{a,d} = \begin{cases} \frac{1}{|\mathcal{T}|} \sum_{\tau \in \mathcal{T}} U_a(\tau), & \text{if } d \text{ is available} \\ -\infty, & \text{otherwise} \end{cases} \tag{5}$$

## Models and model fitting

We adopted a softmax policy to linearly combine the utility values under each basis strategy and used MLE to estimate the model parameters with the monkeys' behavior.

### Utility preprocessing

To combine the strategies and produce a decision, we first preprocess the utility data computed from the decision trees with two steps. First, because two *evade* strategies have negative utility values, we calculate their difference to the worst-case scenario within a trial and use the difference, which is a positive value, as the utility for the two *evade* strategies:

$$U_a^{(t)} = U_a^{(t)} - \min_t \left( U_a^{(t)} \right) \text{ for } a \in \{evade \text{ Blinky}, evade \text{ Clyde}\} \tag{6}$$

Second, because the scale of utility value varies in different strategies, we normalize the utilities within each strategy:

$$U_a^{(t)} = \frac{U_a^{(t)}}{\max\left(U_a^{(t)}\right)} \text{ for } a \in A \tag{7}$$

with $\mathcal{A} = local, global, evade \text{ Blinky}, evade \text{ Clyde}, approach, energizer$.

### Softmax policy

With the adjusted and normalized utility values, each strategy $a$ is associated with a set of utility values $\boldsymbol{U}_{a,d}$ for four directions $d \in \mathcal{D}$. We compute the utility for each direction $d$ by simply combining them linearly with strategy weights $\boldsymbol{w} \in R^6$:

$$Q_d^{(t)} = \sum_{a \in \mathcal{A}} w_a \boldsymbol{U}_{a,d}^{(t)}. \tag{8}$$

The final decision is based on a softmax policy:

$$\pi(d|\boldsymbol{w}) = \frac{\exp\left(Q_d^{(t)}\right)}{\sum_{d' \in \mathcal{D}} \exp\left(Q_{d'}^{(t)}\right)}, \tag{9}$$

where $\pi\left(d|\boldsymbol{w}\right)$ describes the probability of choosing $d$ given weights $\boldsymbol{w}$.

## Maximum likelihood estimate

We use the MLE approach to estimate monkeys' strategy weights $\mathbf{w}$ in a time window $\delta$. Based on Pac-Man's actual moving directions $\boldsymbol{d}^*$, we compute the likelihood as

$$L\left(\boldsymbol{d}^*|w,\delta\right) = \prod_{t\in\delta}\frac{\exp\left(Q_{d*}^{(t)}\right)}{\sum_{d'\in\mathcal{D}}\exp\left(Q_{d'}^{(t)}\right)} \tag{10}$$

The strategy weights within a time window can be estimated by maximizing the log-likelihood:

$$\hat{\boldsymbol{w}} = \underset{\boldsymbol{w}}{\operatorname{argmax}}\sum_{t\in\delta}\left(Q_{d*}^{(t)} - \log\left(\sum_{d'\in\mathcal{D}}\exp(Q_{d'}^{(t)})\right)\right) \tag{11}$$

## Dynamic compositional strategy model

The dynamic compositional strategy model estimates the strategy weights using time windows of flexible length. We assume that the relative strategy weights are stable for a period. The weights can be estimated from the monkeys' choices during this period. We design a two-pass fitting procedure to divide each trial into segments of stable strategies and extract the strategy weights for each segment, avoiding potential overfitting caused by segmentations too fine with too many weight parameters while still capturing the strategy dynamics. The procedure is as follows:

1. We first formulate fine-grained time windows $\Delta = \delta_1, \delta_2, \cdots, \delta_k$ according to the following events: Pac-Man direction changes, ghost consumptions, and energizer consumptions. The assumption is that the strategy changes only occurred at those events.
2. The first-pass fitting is done by using the fine-grained time windows to get the maximum likelihood estimates of the strategy weights as a time series of $w^{\delta_1}, \cdots, w^{\delta_k}$.
3. We then use a change-point detection algorithm to detect any changes in the strategy weights in the time series of $w^{\delta_1}, \cdots, w^{\delta_k}$. Specifically, we select a changing-points number $K$ and used a forward dynamic programming algorithm (*Truong et al., 2020*) to divide the series into $K$ segments $\Delta_K = \delta_1, \delta_2, \cdots, \delta_K$ by minimizing the quadratic loss $c\left(\Delta_K\right) = \sum_{\delta\in\Delta_K}\|w^{\delta} - \bar{w}\|_2^2$. Here, $\bar{w}$ is the empirical mean of these fine-grained weights corresponding to segment sets, $\Delta_K$. With $\Delta_K$, we construct the coarse-grained time windows by combining the fine-grained time windows.
4. The second-pass fitting is then done using the coarse-grained time windows $\Delta_K$ with MLE, and the sum of log-likelihood $L\left(K\right) = \sum_{\delta\in\Delta_K}\log\mathcal{L}\left(\boldsymbol{d}^*|w^{\delta}, \delta\right)$ is the loss function.
5. We repeat the steps 3 and 4 with hyperparameter $K$ traversing through 2, 3, ..., 20 to find out $K^* = \operatorname{argmax}L\left(K\right)$. The final fitting results are based on the normalized fitted weights with $K^*$ coarse-grained time windows $\Delta_{K^*}$.

To ensure that the fitted weights are unique (*Buja et al., 1989*) in each time window, we combine utilities of any strategies that give exactly the same action sequence and reduce multiple strategy terms (e.g., local and energizer) to one hybrid strategy (e.g., local + energizer). After MLE fitting, we divide the fitted weight for this hybrid strategy equally among the strategies that give the same actions in the time segments.

## Static strategy model

The static strategy model uses all data to estimate a single set of strategy weights.

## LARL model

The model shares the same structure with a standard Q-learning algorithm but uses the monkeys' actual joystick movements as the fitting target. To highlight the flatness of the model, we adopt a common assumption that the parameterization of the utility function is linear (*Sutton and Barto, 2018*) with respect to the seven game features: $Q_{\boldsymbol{\theta}}\left(s, d\right) = \sum_i\theta_i \cdot x_i\left(s, d\right)$. These features include the

local pellet number within five steps in four directions $x_{local}$ , the Dijkstra distance to the closest pellet $x_{closest}$ , the Dijkstra distance to the closest energizer $x_e$ , the global pellet number (distance larger than five steps) weighted by their inverse Dijkstra distances to Pac-Man $x_{global}$ , the Dijkstra distance to Blinky $x_{gB}$ , the Dijkstra distances to Clyde $x_{gC}$, and the Dijkstra distance to the closest scared ghost $x_{gS}$ . If a feature is not available in some game context (e.g., $x_{gS}$ is not available when ghosts are in the normal mode or the dead mode), we denote it to be null. The update rule follows the standard temporal-difference learning rule:

$$\theta'_i = \theta_i + \alpha \left( r\left(s, d, s'\right) + \gamma Q_{\boldsymbol{\theta}}\left(s', d'\right) - Q_{\boldsymbol{\theta}}\left(s, d\right) \right) \cdot x_i\left(s, d\right) \tag{12}$$

where $\alpha$ is the learning rate, $\gamma$ is the discount factor, and $r\left(s, d, s'\right)$ is the reward that the agent received from state $s$ to $s$ via action $d$. All the reward values are the actual rewards that the monkeys received in the game. Compared to a typical TD update rule, the max operation in the utility-to-go term $r\left(s, d, s'\right) + \gamma\ max_{d'}\ Q_{\boldsymbol{\theta}}\left(s', d'\right)$ is replaced with the utility under the monkeys' actual joystick movement $r\left(s, d, s'\right) + \gamma\ Q_{\boldsymbol{\theta}}\left(s', d'\right)$ . Feature weights are randomly initialized, and we use the monkey behavioral data to update these weights. There are two model hyper-parameters: learning rate $\alpha$ and discount factor $\gamma$. They are selected through threefold cross-validation. The best hyperparameters are $\alpha = 0.01$, $\gamma = 0.3$ for Monkey O and $\alpha = 0.025$, $\gamma = 0.3$ for Monkey P. The trained feature weights $\boldsymbol{\theta}$ are shown in *Appendix 1—table 7*.

## Linear perceptron model

We build a linear perceptron as another representative flat descriptive model (without calculating utilities and strategies) to describe monkeys' decision-making based on the same 20 features included in the other models. These features include the modes of Blinky and Clyde (two features), Dijkstra distances between Blinky and Pac-Man in four directions (four features), Dijkstra distances between Clyde and Pac-Man in four directions (four features), Dijkstra distances between the closest energizer and Pac-Man in four directions (four features), distances between fruits and Pac-Man in four directions (four features), the number of pellets within 10 steps of Pac-Man, and the number of pellets left in the maze. For unavailable directions, the corresponding feature value is filled with a none value. We trained a three-layer perceptron with monkeys' choice behavior: an input layer for 20 features, a hidden layer, and an output layer for four directions. We used scikit-learn (https://scikit-learn.org/). The size of the hidden layer was selected from $n_{hidden} =$\{16, 32, 64, 128, 256\} with the largest average prediction accuracy on all data. For Monkey O, the best hidden unit number is 64, and for Monkey P, the best hidden unit number is 128. Each model uses Adam for optimization, training batch size = 128, learning rate = 0.001, regularization parameter = 0.0001, and the activation function $f\left(\cdot\right)$ for the hidden layer is an identity function.

## Model comparison

We compare four models (static strategy model, dynamic strategy model, LARL, and linear perceptron model) in four game contexts shown in *Figure 2B*, *Figure 2—figure supplement 1A and E*, and *Appendix 1—table 4*. Fivefold cross-validations are used to evaluate the fitting performance of these models with each monkeys' behavior data.

## Strategy heuristic analysis

We label the behavior strategy as *vague* when the weight difference between the largest and the second largest strategies is less than 0.1 (*Figure 2D*). Otherwise, the labels are based on the strategy with the largest weight.

In *Figure 3A and B*, *Figure 3—figure supplement 1A, B, E, and F*, we evaluate the strategy probability dynamics with respect to two features: local pellet density (the number of pellets within 10 steps from Pac-Man) and scared ghost distance. We group the behavior data based on these two features and calculate the frequency of the relevant strategies in each. Means and standard deviations are computed by bootstrapping 10 times with a sample size of 100 for each data point (*Figure 3A and B*, *Figure 3—figure supplement 1A, B, E and F*).

In *Figure 3C and D* and *Figure 3—figure supplement 1C, D, G and H*, monkeys' moving trajectories with at least four consecutive steps labeled as *global* strategy are selected. We use Dijkstra's

algorithm to compute the shortest path from the starting position when the monkey switched to *global* strategy to the ending position when the monkey first reached a pellet. The trajectory with the fewest turns is determined by sorting all possible paths between the starting and the ending position.

## Eye movement analysis

We label monkeys' fixation targets based on the distance between the eye position and the relevant game objects: Pac-Man, ghosts, pellets, and energizer. When the distances are within one tile (25 pixel), we add the corresponding target to the label. There can be multiple fixation labels because these objects may be close to each other.

In *Figure 3E* and *Figure 3—figure supplement 2A and E*, we select strategy periods with more than 10 consecutive steps and compute the fixation ratio by dividing the time that the monkeys spent looking at an object within each period by the period length. As pellets and energizers do not move but Pac-Man and ghosts do, we do not differentiate between fixations and smooth pursuits when measuring where the monkeys looked at. We compute the average fixation ratio across the periods with the same strategies.

In the pupil dilation analyses in *Figure 3F* and *Figure 3—figure supplement 2B, C, D, F, G, and H*, we z-score the pupil sizes in each game round. Data points that are three standard deviations away from the mean are excluded. We align the pupil size to strategy transitions and calculate the mean and the standard error (*Figure 3F*, solid line and shades). As the control, we select strategy transitions that go through the *vague* strategy and align the data to the center of the vague period to calculate the average pupil size and the standard error (*Figure 3F*, *Figure 3—figure supplement 2B and F*, dashed line and shades). *Figure 3—figure supplement 2C, D, G and H* are plotted in a similar way but only specific strategy transitions.

## Compound strategy analysis

### Planned attack

We define *planned attack* and *accidental consumption* trials according to the strategy labels after the energizer consumption: when at least 8 out of the 10 time steps after the energizer consumption are labeled as the *approach* strategy, the trial is defined as *planned attack*; otherwise, this trial is defined as *accidental consumption*. There are 478 (Monkey O) and 459 (Monkey P) *planned attack* trials and 1984 (Monkey O) and 1257 (Monkey P) *accidental consumption* trials. These trials are aligned to the time of energizer consumption in *Figure 5A* and *Figure 5—figure supplement 1A and E*.

In *Figure 5B* and *Figure 5—figure supplement 1B and F*, the average of Pac-Man-energizer distance, energizer-ghost distance, and Pac-Man-ghost distance is computed at the beginning of the *planned attack* and *accidental consumption* trials. The beginning of each trial is defined as the position where Pac-Man started to take the direct shortest route toward the energizer. The average fixation ratios in *Figure 5C* and *Figure 5—figure supplement 1C and G* are computed from the beginning of each *planned attack* or *accidental consumption* till when the energizer is eaten.

In some of the *accidental consumption* trials (Monkey O: 625/31.5%; Monkey P: 477/37.9%), Pac-Man caught a ghost although the monkeys did not pursue the ghosts immediately after the energizer consumption. In contrast, all *planned attack* trials resulted in Pac-Man catching the ghosts successfully. These trials are aligned to the ghost consumption in *Figure 5D* and *Figure 5—figure supplement 1D and H*.

### Suicide

We define *suicide* and *failed evasion* trials based on the strategy labels in the last ten steps before Pac-Man's death: a trial is defined as *suicide* when all 10 steps are labeled as *approach* and as *failed evasion* when all steps are labeled as *evade*.

In *Figure 6A* and *Figure 6—figure supplement 1A and E*, the distance between Pac-Man and the closest pellet and the distance between Pac-Man reset location and the ghost are computed at the time point when the monkeys switched to the *approach* (*suicide*) or *evade* (*failed evasion*) strategy. Also, in *Figure 6B* and *Figure 6—figure supplement 1B and F*, the average distance between Pac-Man and two ghosts is computed in the same condition. The average fixation ratios in *Figure 6C* and *Figure 6—figure supplement 1C and G* are computed from that time point until Pac-Man's death. In

*Figure 6D* and *Figure 6—figure supplement 1D and H*, the relative pupil sizes are aligned to Pac-Man's death.

## Acknowledgements

We thank Wei Kong, Lu Yu, Ruixin Su, Yunxian Bai, Zhewei Zhang, Yang Xie, Tian Qiu, Yiwen Xu, Ce Ma, Zhihua Zhu, and Yue Hao for their help in all phases of the study, and Liping Wang and Xaq Pitkow for providing comments and advice.

## Additional information

### Funding

| Funder | Grant reference number | Author |
|---|---|---|
| National Science and Technology Innovation 2030 Major Program | 2021ZD0203701 | Tianming Yang |
| Chinese Academy of Sciences | XDB32070100 | Tianming Yang |
| Shanghai Municipal Science and Technology Major Project | 2018SHZDZX05 | Tianming Yang |
| National Natural Science Foundation of China | 32100832 | Qianli Yang |

The funders had no role in study design, data collection and interpretation, or the decision to submit the work for publication.

### Author contributions

Qianli Yang, Data curation, Formal analysis, Funding acquisition, Investigation, Methodology, Validation, Visualization, Writing – original draft, Writing – review and editing; Zhongqiao Lin, Formal analysis, Investigation, Methodology, Software, Writing – review and editing; Wenyi Zhang, Jianshu Li, Data curation, Formal analysis, Investigation; Xiyuan Chen, Data curation, Investigation; Jiaqi Zhang, Formal analysis; Tianming Yang, Conceptualization, Formal analysis, Funding acquisition, Investigation, Methodology, Project administration, Resources, Software, Supervision, Validation, Visualization, Writing – original draft, Writing – review and editing

### Author ORCIDs

Qianli Yang (iD) http://orcid.org/0000-0003-4226-2319
Jiaqi Zhang (iD) http://orcid.org/0000-0002-1649-3378
Tianming Yang (iD) http://orcid.org/0000-0001-6976-9246

### Ethics

All procedures followed the protocol approved by the Animal Care Committee of Shanghai Institutes for Biological Sciences, Chinese Academy of Sciences (CEBSIT-2021004).

### Decision letter and Author response

Decision letter https://doi.org/10.7554/eLife.74500.sa1
Author response https://doi.org/10.7554/eLife.74500.sa2

## Additional files

### Supplementary files
• Transparent reporting form

## Data availability

The data and codes that support the findings of this study are provided at: https://github.com/ superr90/Monkey_PacMan, (copy archived at swh:1:rev:6f74eef3b321718ac2f8d4d4f5f1d904b1 0d2a85).

The following dataset was generated:

| Author(s) | Year | Dataset title | Dataset URL | Database and Identifier |
| --- | --- | --- | --- | --- |
| Yang Q | 2022 | Hierarchical Decision-Making Analysis for Behavior Data | https://github.com/ superr90/Monkey_ PacMan | GitHub, GitHub |

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

## Appendix 1

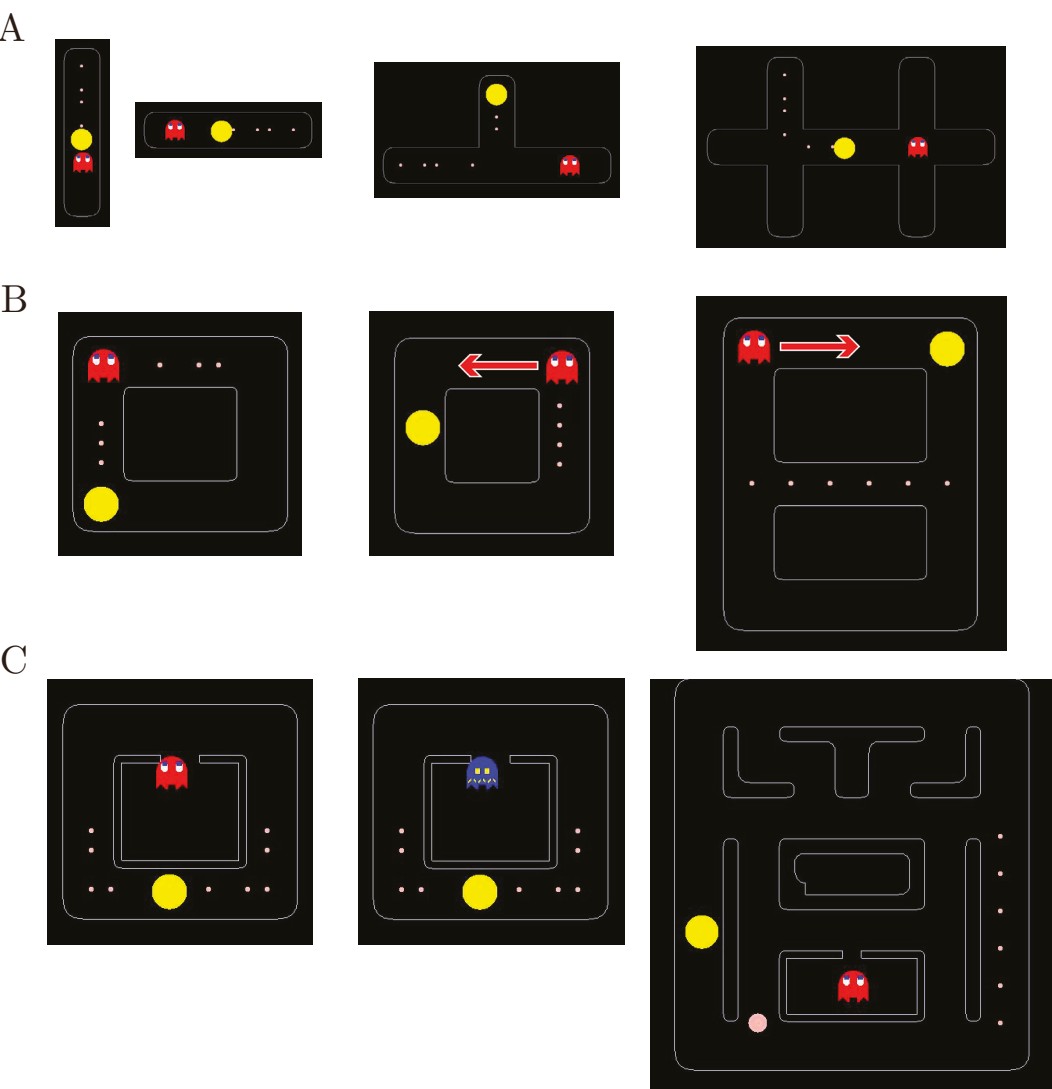

**Appendix 1—figure 1.** Training procedure. (**A**) Stage 1 training mazes. From left to right: (1) Vertical maze. Pac-Man started from the middle position, with several pellets in one direction and a static ghost in the other. The monkeys learned to move the joystick upward and downward. (2) Horizontal maze. The monkeys learned to move the joystick toward left and right. (3) T-maze. Pac-Man started from the vertical arm, and the monkeys learned to move out of it by turning left or right. Pellets were placed in one arm and a static ghost in the other. (4) H-maze. Pac-Man started from the middle of the maze. There were pellets placed on the way leading to one of the three arms, and a static ghost was placed at the crossroad on the opposite side. (**B**) Stage 2 training mazes. From left to right: (1) Square maze with a static ghost. Pac-Man started from one of the four corners, and pellets were placed in two adjacent sides with a static ghost placed at the corner connecting the two. (2) Square maze with a moving ghost. Pac-Man started from the middle of one of the four sides, and pellets were placed on the opposite side. A ghost moved from one end of the pellet side and stopped at the other end. (3) Eight-shaped maze with a moving ghost. Pac-Man stated from one of the four corners. The pellets were placed in the middle tunnel. A ghost started from a corner and moved toward the pellets. (**C**) Stage 3 training mazes. From left to right: (1) Square maze with Blinky. Pac-Man started from the middle of the bottom side with pellets placed on both sides. Blinky in normal mode started from its home. (2) Square maze with a ghost in a permanent scared mode. The scared ghost started from its home. Once caught by Pac-Man, the ghost went back to its home. (3) Maze with an energizer and Blinky. An energizer was randomly placed in the maze. Once the energizer was eaten, the ghost would be turned into the scared mode. The scared mode lasted until the ghost was eaten by Pac-Man. Once the ghost was eaten, it returned to its home immediately and came out again in the normal mode. After the monkeys were able to perform the task, we limited the scared mode to 14 s.

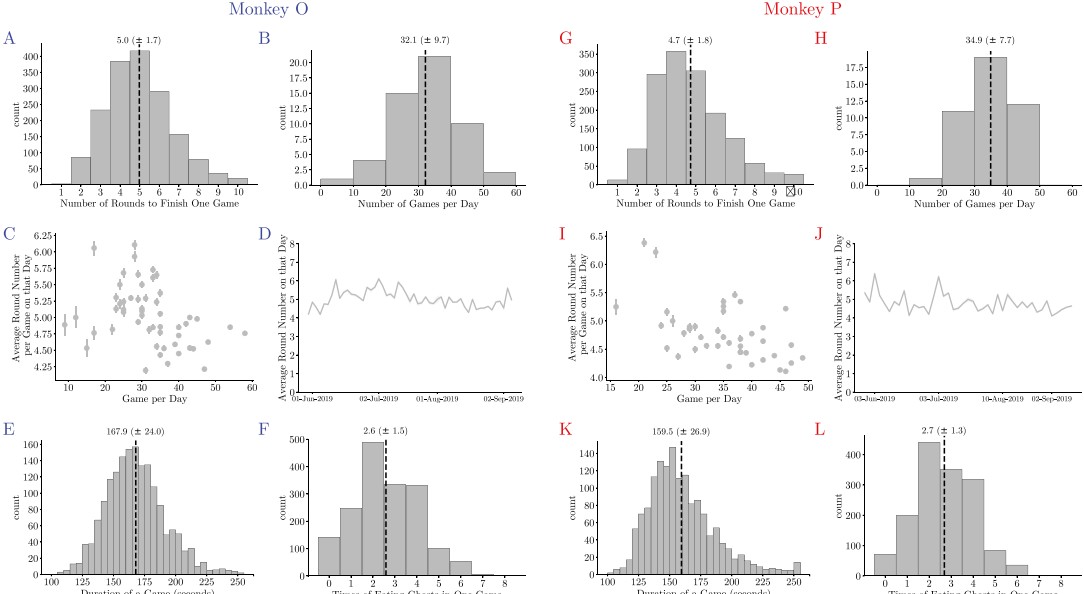

**Appendix 1—figure 2.** Basic game statistics of Monkey O (left) and Monkey P (right). (**A**, **G**) The number of rounds to clear all pellets in each game. Vertical dashed lines denote means. (**B**, **H**) The number of games accomplished on each day. Vertical dashed lines denote means. (**C**, **I**) The average number of rounds to clear a maze plotted against the number of games in a session. Vertical lines denote standard deviations. Playing more games in each session can slightly improve the monkey's game performance. (**D**, **J**) The average number of rounds during the training. (**E**, **K**) The time needed to clear a maze. Vertical dashed lines denote means. (**E**, **K**) The time needed to clear a maze. Vertical dashed lines denote means.

## Pearson Correlation of Strategy Action Sequence

|            | global | local | evade(B) | evade(C) | approach | energizer | random |
|------------|--------|-------|----------|----------|----------|-----------|--------|
| global     | 1.00   | 0.55  | 0.25     | 0.15     | 0.30     | 0.17      | 0.59   |
| local      | 0.55   | 1.00  | 0.15     | 0.05     | 0.16     | 0.26      | 0.49   |
| evade(B)   | 0.25   | 0.15  | 1.00     | 0.49     | 0.55     | 0.01      | 0.26   |
| evade(C)   | 0.15   | 0.05  | 0.49     | 1.00     | 0.37     | -0.00     | 0.18   |
| approach   | 0.30   | 0.16  | 0.55     | 0.37     | 1.00     | 0.01      | 0.30   |
| energizer  | 0.17   | 0.26  | 0.01     | -0.00    | 0.01     | 1.00      | 0.15   |
| random     | 0.59   | 0.49  | 0.26     | 0.18     | 0.30     | 0.15      | 1.00   |

**Appendix 1—figure 3.** Strategy basis correlation matrix. We computed the Pearson correlations between the action sequences chosen with each basis strategy within each coarse-grained segment determined by the two-pass fitting procedure. As a control, we computed the correlation between each basis strategy and a random strategy, which generates action randomly, as a baseline. Most strategy pairs' correlation was lower than the random baseline.

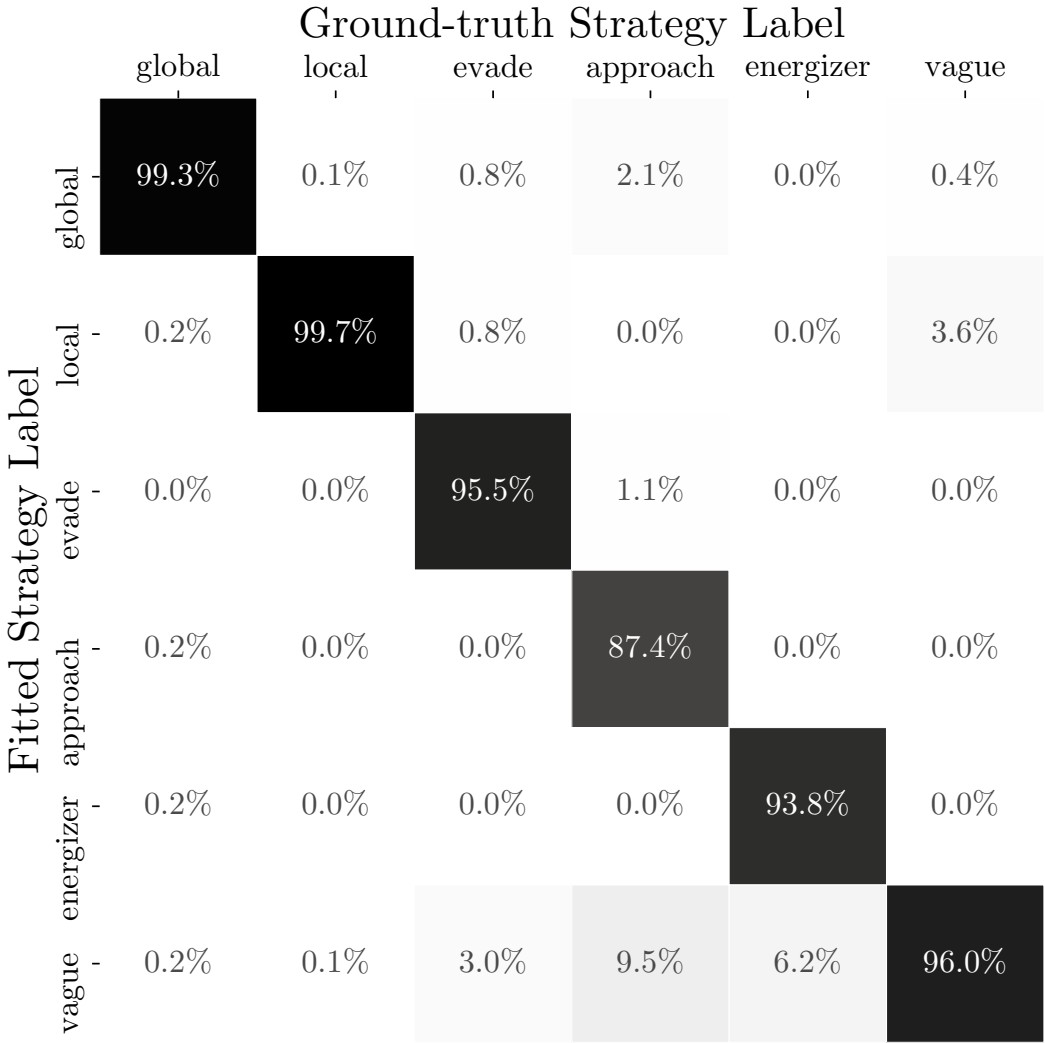

**Appendix 1—figure 4.** Recovering the strategy labels of an artificial agent with the dynamic compositional strategy model based on simulated gameplay data. The confusion matrix between the fitted strategy labels and the ground-truth strategy labels from an artificial agent is shown. The artificial agent used time-varying strategy weights to combine six strategies illustrated in the method. Strategy weights were selected based on two monkeys' choices at the same game context determined by the location and state of Pac-Man, the pellets, the energizers, and the ghosts. We used the dynamic compositional strategy model to estimate the strategy labels from 2050 rounds of simulated data and produced the confusion matrix. In most cases, the model was able to recover the correct strategy (diagonal boxes).

**Appendix 1—table 1.** Four path types in the maze.

| Path type | Selection criteria |
|---|---|
| Straight | Contains two opposite moving directions |
| L-shape | Contains two orthogonal moving directions |
| Fork | Contains three moving directions |
| Cross | Contains four moving directions |

**Appendix 1—table 2.** Awarded and penalized utilities for each game element in the model.

| Reward ($r_p$) | Reward ($r_e$) | Reward ($r_f$)(1–5) | Reward ($g$) | Penalty ($g$) |
|---|---|---|---|---|
| 2 | 4 | 3, 5, 8, 12, 17 | 8 | -8 |

**Appendix 1—table 3.** Special game contexts and corresponding selection criteria.

| Context | Selection criteria |
|---|---|
| All stage | n.a. |
| Early game | Remaining number of pellets $\geq$ 90% of total (80 for O and 65 for P) |
| Late game | Remaining number of pellets $\leq$ 10% of total (10 for O and 7 for P) |
| Scared ghost | Any scared ghosts within 10 steps away from Pac-Man |

**Appendix 1—table 4.** Comparison of prediction accuracy (± SE) across four models in four game contexts for the two monkeys.

| | Strategy | | | |
|---|---|---|---|---|
| Context | Dynamic | Static | LARL | Perceptron |
| Overall | 0.904 ± 0.006 | 0.816 ± 0.010 | 0.669 ± 0.011 | 0.624 ± 0.010 |
| Early game | 0.886 ± 0.0016 | 0.804 ± 0.025 | 0.775 ± 0.021 | 0.582 ± 0.026 |
| Late game | 0.898 ± 0.011 | 0.805 ± 0.019 | 0.621 ± 0.018 | 0.599 ± 0.019 |
| Scared ghosts | 0.958 ± 0.010 | 0.728 ± 0.031 | 0.672 ± 0.025 | 0.455 ± 0.030 |

LARL: linear approximate reinforcement learning.

**Appendix 1—table 5.** Comparison of prediction accuracy (± SE) across four models in four game contexts for Monkey O.

| | Strategy | | | |
|---|---|---|---|---|
| Context | Dynamic | Static | LARL | Perceptron |
| Overall | 0.907 ± 0.008 | 0.806 ± 0.014 | 0.659 ± 0.016 | 0.632 ± 0.013 |
| Early game | 0.868 ± 0.0132 | 0.772 ± 0.054 | 0.765 ± 0.042 | 0.548 ± 0.037 |
| Late game | 0.901 ± 0.016 | 0.786 ± 0.027 | 0.595 ± 0.026 | 0.598 ± 0.026 |
| Scared ghosts | 0.952 ± 0.019 | 0.729 ± 0.047 | 0.658 ± 0.046 | 0.545 ± 0.050 |

LARL: linear approximate reinforcement learning.

**Appendix 1—table 6.** Comparison of prediction accuracy (± SE) across four models in four game contexts for Monkey P.

| | Strategy | | | |
|---|---|---|---|---|
| Context | Dynamic | Static | LARL | Perceptron |
| Overall | 0.900 ± 0.009 | 0.825 ± 0.012 | 0.679 ± 0.015 | 0.615 ± 0.014 |
| Early game | 0.898 ± 0.017 | 0.824 ± 0.021 | 0.781 ± 0.022 | 0.605 ± 0.035 |
| Late game | 0.894 ± 0.016 | 0.826 ± 0.026 | 0.653 ± 0.025 | 0.599 ± 0.028 |
| Scared ghosts | 0.962 ± 0.011 | 0.723 ± 0.041 | 0.682 ± 0.027 | 0.456 ± 0.037 |

LARL: linear approximate reinforcement learning.

**Appendix 1—table 7.** Trained feature weights in the LARL model for Monkey O and Monkey P.

| Monkey | Feature $x_{gB}$ | $x_{gC}$ | $x_{gS}$ | $x_{local}$ | $x_{closest}$ | $x_{global}$ | $x_e$ |
|--------|-------|-------|---------|---------|-----------|----------|---------|
| O | 0.3192 | 0.0049 | −0.0758 | 2.4743 | −0.7799 | 1.0287 | −0.9717 |
| P | 0.5584 | 0.0063 | −0.0563 | 2.5802 | −0.6324 | 1.6068 | −1.1994 |

LARL: linear approximate reinforcement learning.

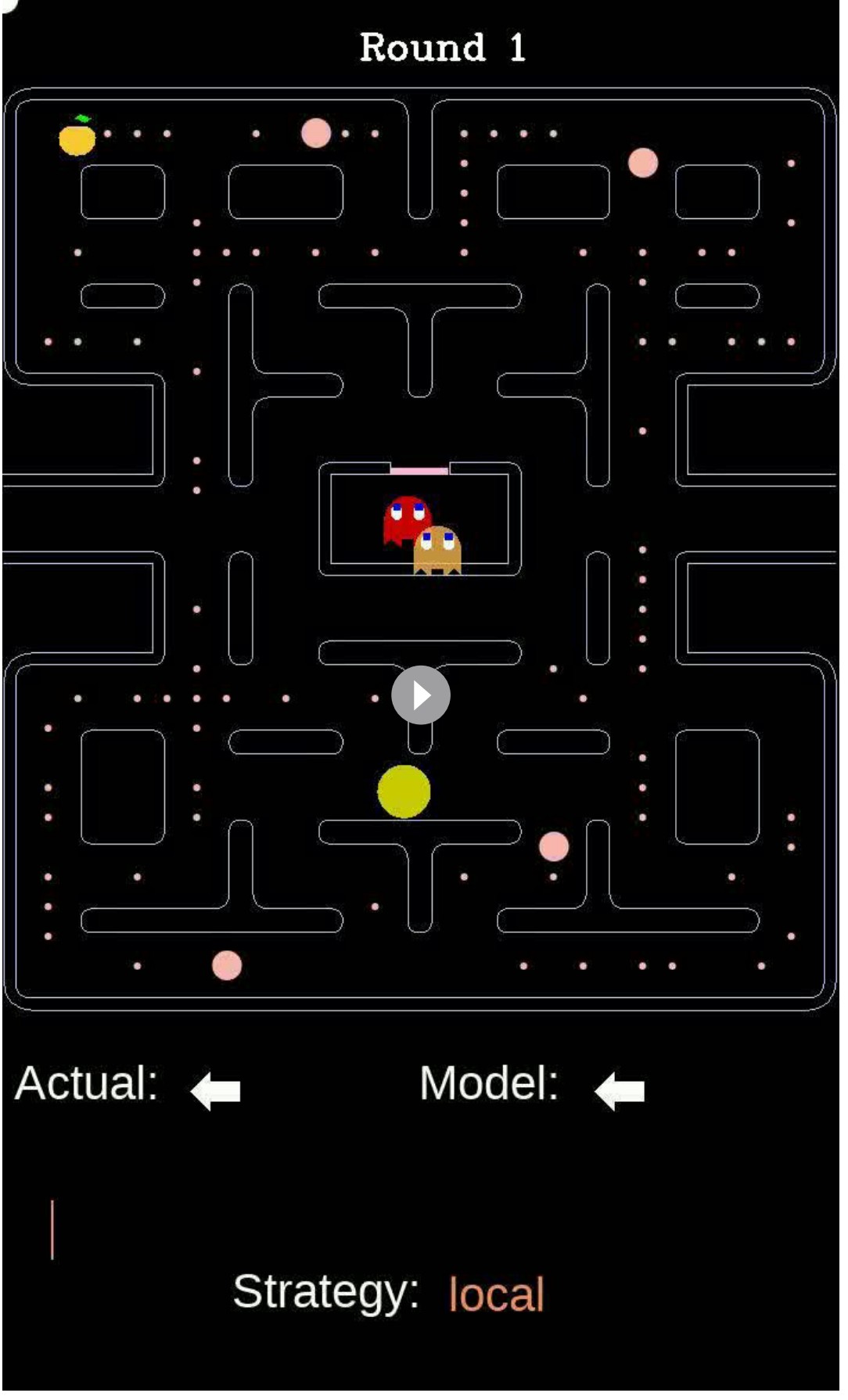

**Appendix 1—video 1.** Example game trials. Monkey O's moving trajectory, actual and predicted actions, and labeled strategies are plotted in these example game trials.

https://elifesciences.org/articles/74500/figures#video1

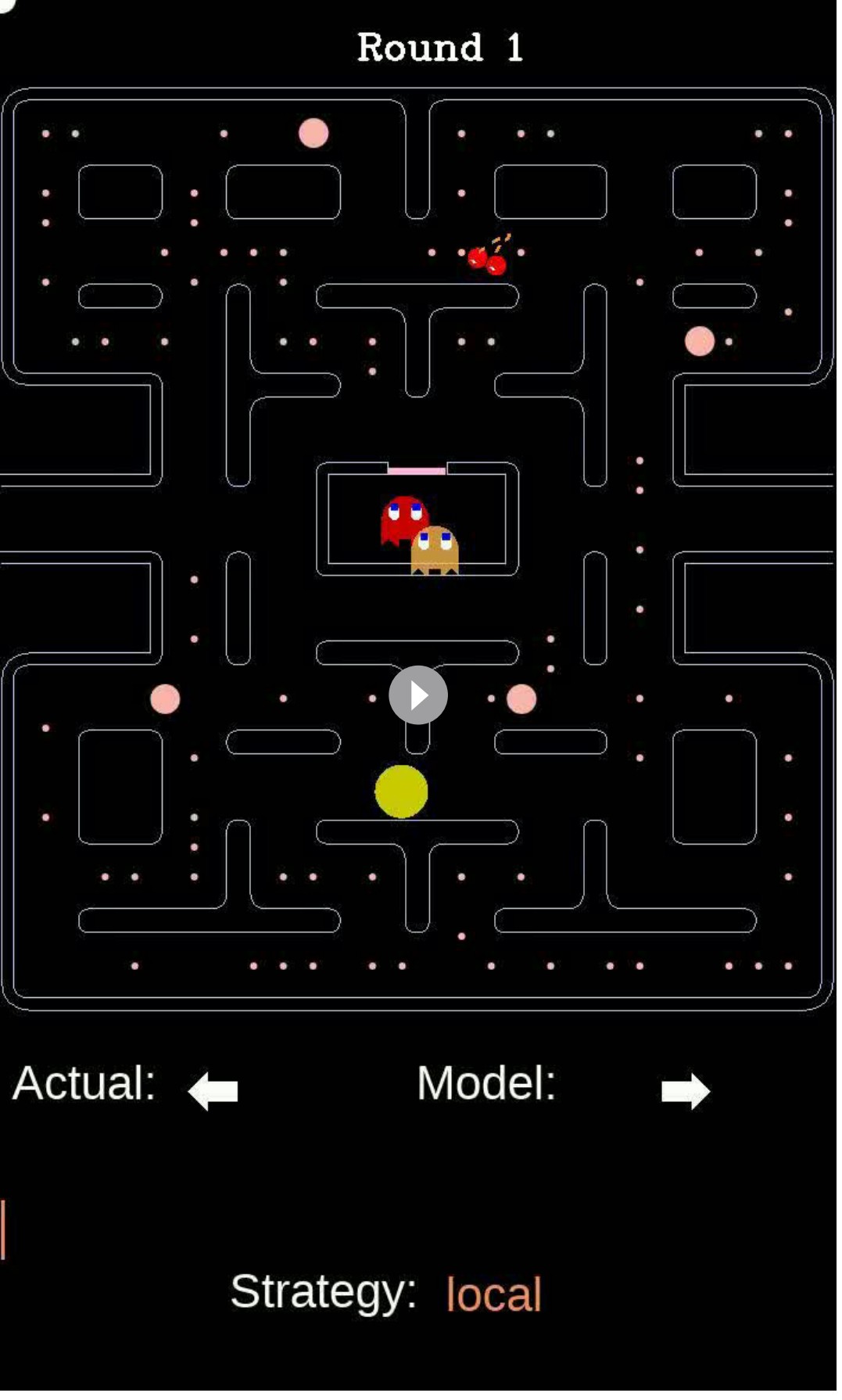

**Appendix 1—video 2.** Example game trials. Monkey O's moving trajectory, actual and predicted actions, and labeled strategies are plotted in these example game trials.

https://elifesciences.org/articles/74500/figures#video2

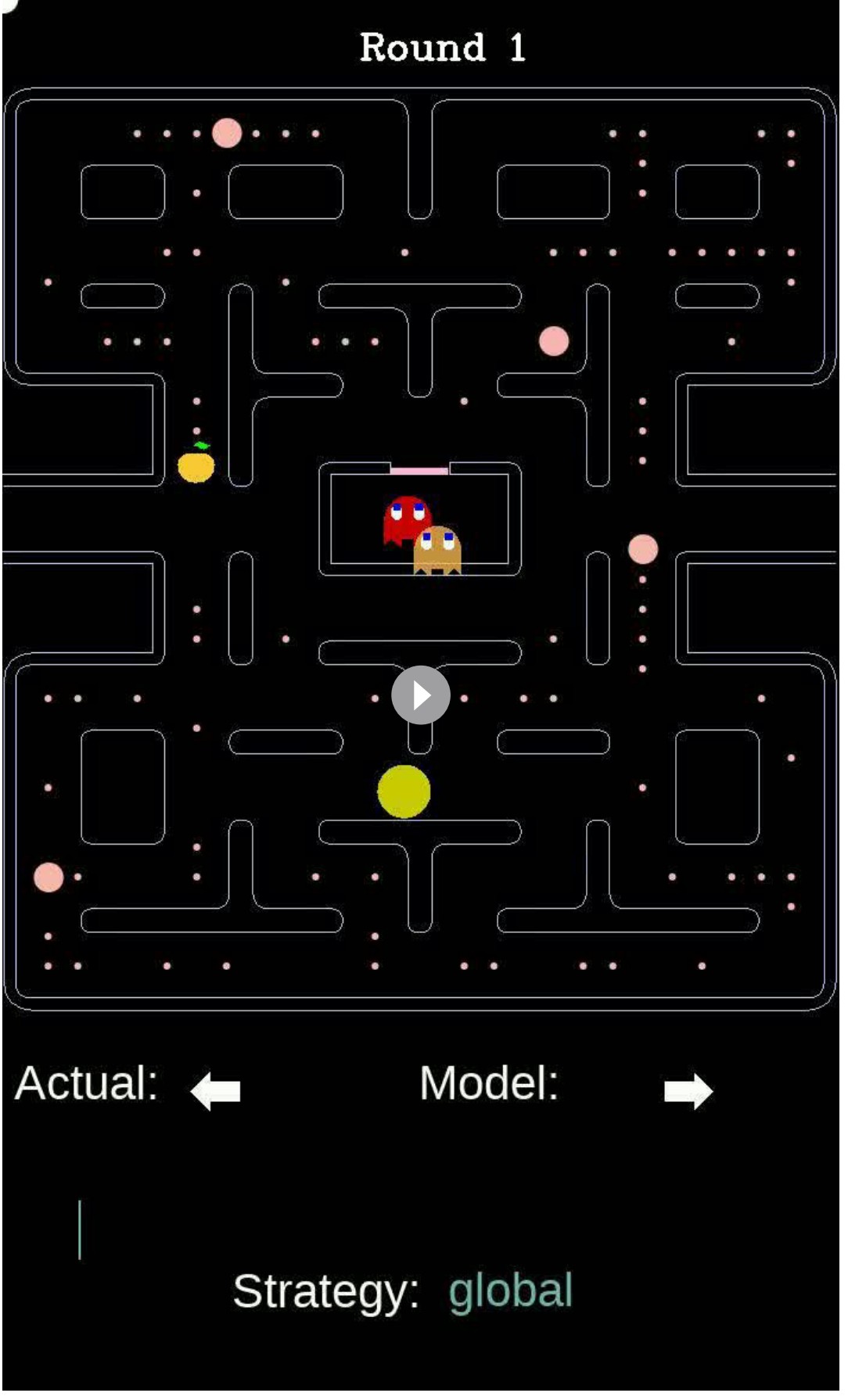

**Appendix 1—video 3.** Example game trials. Monkey O's moving trajectory, actual and predicted actions, and labeled strategies are plotted in these example game trials.

https://elifesciences.org/articles/74500/figures#video3

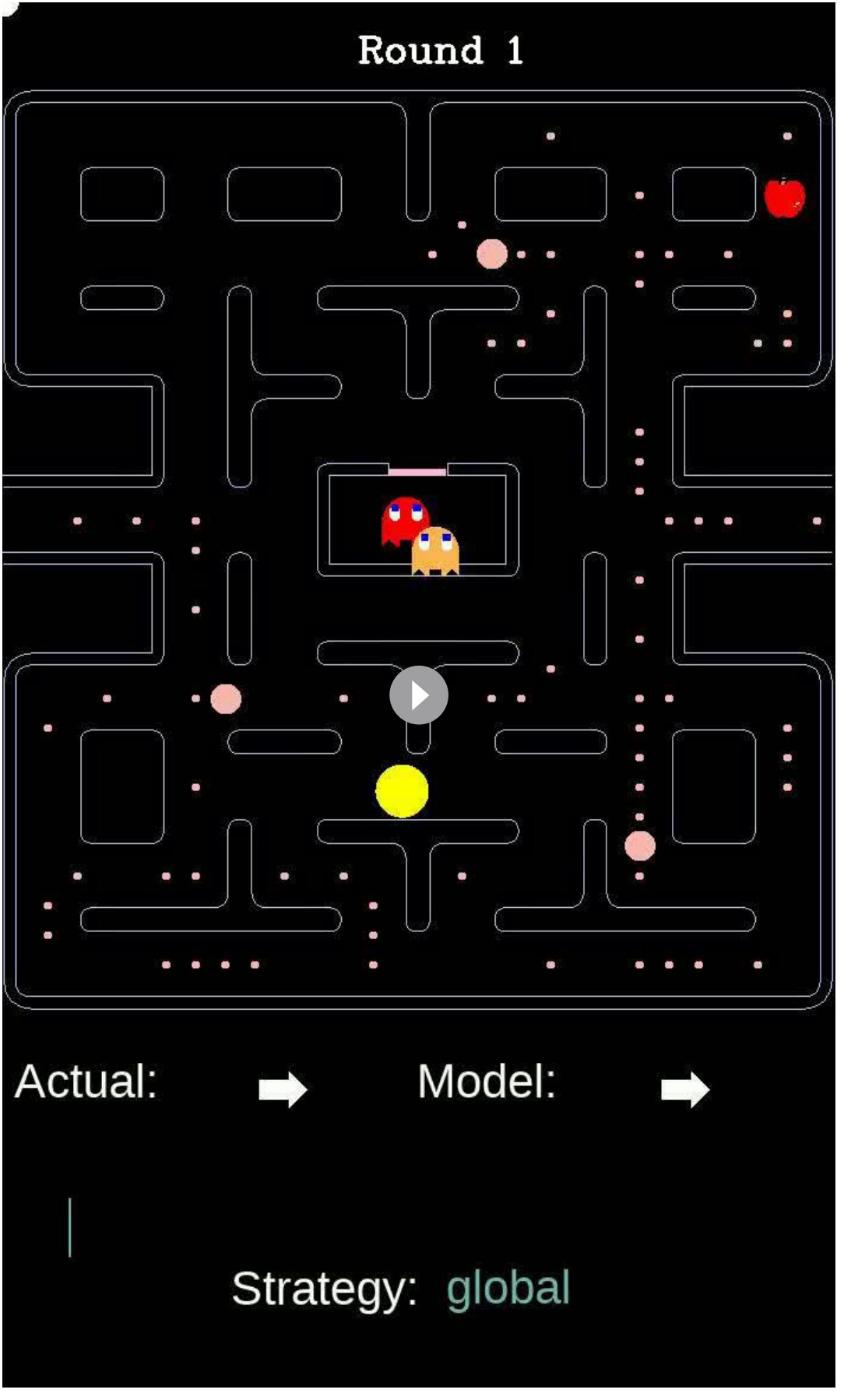

**Appendix 1—video 4.** Example game trials. Monkey P's moving trajectory, actual and predicted actions, and labeled strategies are plotted in this example game segment.

https://elifesciences.org/articles/74500/figures#video4

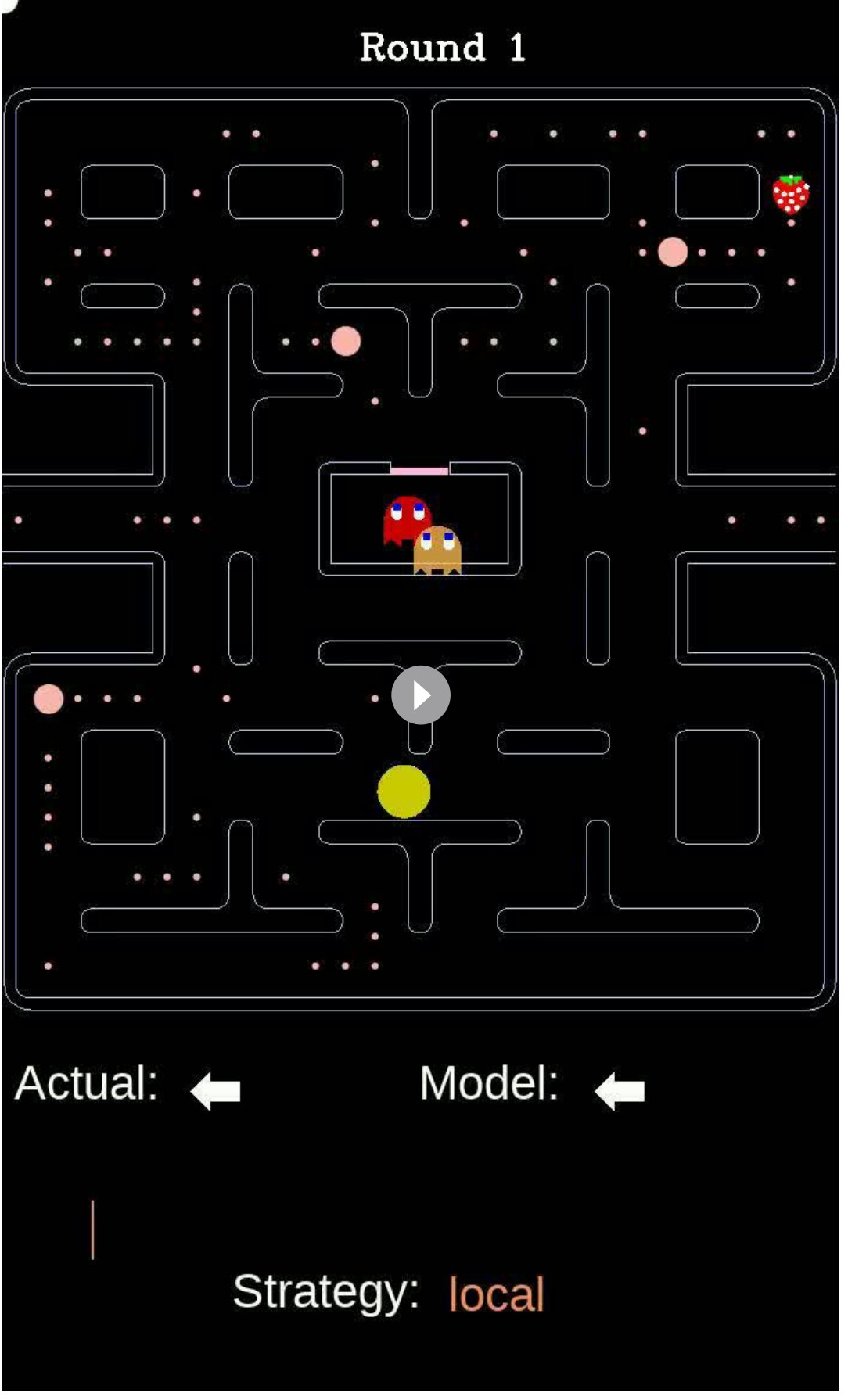

**Appendix 1—video 5.** Example game trials. Monkey P's moving trajectory, actual and predicted actions, and labeled strategies are plotted in this example game segment.

https://elifesciences.org/articles/74500/figures#video5

