## [Editor Report]

Dr. Yang and colleagues trained nonhuman primates (rhesus monkeys) to play a semi-controlled version of the video game Pac-Man. This novel experimental paradigm allowed the authors to analyze and model the kinds of heuristic behavioral strategies monkeys use to solve relatively complex problems. The results provide insight into higher cognition in primates.

---

## [Decision Letter]

**Decision letter after peer review:**

Thank you for submitting your article "Monkey Plays Pac-Man with Compositional Strategies and Hierarchical Decision-making" for consideration by *eLife*. Your article has been reviewed by 3 peer reviewers, and the evaluation has been overseen by a Reviewing Editor and Joshua Gold as the Senior Editor. The following individual involved in review of your submission has agreed to reveal their identity: Bruno B Averbeck (Reviewer #1).

Essential revisions:

The authors report on a complex behavior paradigm together with sophisticated modeling to study higher cognition in primates. The key claims of the manuscript are mostly supported by the data, and the approaches are rigorous. The reviewers were overall enthusiastic about this work, yet identified three key ways in which the findings should be clarified and the report improved:

1) An identified weakness was a lack of comparison with other models and a need for better justification of the modeling strategy. This is especially important since the dataset is very complex. Rather than solely a quantitative comparison of model performance, which may dilute the argument, there could be some discussion of an account for non-hierarchical possibilities. For example, a large neural network (i.e., a non-hierarchical model) could be trained on these trials and produce equal performance or better prediction. The authors should in some way provide more of a rationale for their approach. This could be in discussion, presentation of converging evidence, and/or additional model comparison.

2) Also related to #1, the reviewers suggested that perhaps a way to better justify the choice of model would be to do a selective comparison within the data (i.e. one or two better-chosen baselines) to support the hierarchical claim.

3) The authors should acknowledge that monkeys do not always perform this task in the optimal way, and that monkeys perhaps use a more passive strategy, or use multiple strategies at the same time. The monkeys could obtain almost all the rewards by the end of the game as there is nothing to pressure or force them to optimize their choices. Authors should more fully account for this feature in the experimental design when interpreting their data.

*Reviewer #1 (Recommendations for the authors):*

1. What do you mean by saying "a reasonable level" (line 114)? You may want to show the criteria.

2. Although Clyde tends to keep a certain distance from Pac-Man, why does Pac-Man also chase it when they are close to each other? (figure 1C)

3. Essential references are needed when explaining the behavior or mentioning previous literature, for example, in lines 316 and 485.

4. More details about how you tweaked the task are needed, such as how the monkeys operated the joystick? Holding it to move the Pac-Man, I guess.

*Reviewer #2 (Recommendations for the authors):*

– I think there are potential problems with identifiability in the model as detailed. If, for instance, the authors used a model with known time-varying weights to play the game and then fit their model to these data, do they correctly recover the weights?

– I am unfamiliar with the sense of "hierarchical" behavioral models as used here. By contrast, hierarchical models in RL involve goals, subgoals, etc., with models nested inside one another. If the authors are borrowing this less common sense of the term from somewhere, it would be helpful to cite it. I am not sure in what sense a perceptron is a "flat" model.

– Figure 1B: I find it hard to distinguish the grayscale lines and match them with their corresponding choice types. Also, I don't see standard errors here, as indicated in the caption.

– Figure 2: I found the boxes around the figure panels visually unappealing. This occurs in a couple of other figures.

– Figure 2C-D: If I follow, these are violin plots, not histograms, as stated in the caption.

– Figure 2E: The authors might consider a radar chart as an alternative way of representing these proportion data. Such a chart would allow them to plot each column here as one line and might facilitate more direct comparisons across strategy proportions.

– ll. 206-210: Before having read the methods, this text does not make it clear to me what the authors have done. That is, I don't know what it means here to linearly combine basis strategies or to "pool predictions." More specific language would be helpful, even if just a few words.

– Figure 3C-D: I don't think the dotted magenta lines are necessary, and they make the numbers hard to read.

– I was not sure what to take away from Figure 4. As a conceptual description, perhaps it belongs earlier in the text? Perhaps with some schematic as to how models are combined?

– ll 748-749: Shouldn't utilities for impossible actions be -∞? That is, in any probabilistic action selection model, there will be a probability of selecting any action with a *finite* utility difference, so the difference would need to be infinite to rule out an action.

– ll. 918-920: Three standard deviations seems like a pretty drastic cutoff for outliers. Why not five or at least four? Does this make a difference?

---

## [Author Response]

Essential revisions:The authors report on a complex behavior paradigm together with sophisticated modeling to study higher cognition in primates. The key claims of the manuscript are mostly supported by the data, and the approaches are rigorous. The reviewers were overall enthusiastic about this work, yet identified three key ways in which the findings should be clarified and the report improved:1) An identified weakness was a lack of comparison with other models and a need for better justification of the modeling strategy. This is especially important since the dataset is very complex. Rather than solely a quantitative comparison of model performance, which may dilute the argument, there could be some discussion of an account for non-hierarchical possibilities. For example, a large neural network (i.e., a non-hierarchical model) could be trained on these trials and produce equal performance or better prediction. The authors should in some way provide more of a rationale for their approach. This could be in discussion, presentation of converging evidence, and/or additional model comparison.2) Also related to #1, the reviewers suggested that perhaps a way to better justify the choice of model would be to do a selective comparison within the data (i.e. one or two better-chosen baselines) to support the hierarchical claim.

In the frontier of Machine Learning and Reinforcement Learning, hierarchical and flat models are differentiated by the action levels they are modeling (Sutton and Barto, 2018). Flat models find the policy based on the optimal primitive actions (e.g., joystick movements) at each step. For example, a typical flat model may use a standard Q-learning algorithm. In contrast, hierarchical models make goal-directed and temporally extended high-level decisions based on which primitive decisions are made. Optimal policies are searched over these high-level decisions. These high-level decisions are the *strategies* in our paper but may have different names in the literature, such as *skills*, *operators*, *macro-operators*, *macro-actions*, or *options* (M. M. Botvinick et al., 2009; M. Botvinick and Weinstein, 2014; Sutton et al., 1999).

Based on the distinction between hierarchical and flat models, we did the following in the new revision to argue that the monkeys used a hierarchical model instead of a flat model to play the Pac-Man game.

Firstly, in Results, we include a linear approximate reinforcement learning model (LARL) (Sutton, 1988; Tsitsiklis and Van Roy, 1997). The LARL model shared the same structure with a standard Q-learning algorithm but used the monkeys’ actual joystick movements as the fitting target. To highlight the flatness of this baseline model, we adopted a common assumption that the parameterization of the utility function is linear (Sutton and Barto, 2018) with respect to 7 game features. In updated Figure 2B, Figure 2—figure supplement 1AE, and Appendix Table 4-6, we compare the prediction accuracy of this model in four different game contexts. The LARL model only achieves 0.669±0.011 overall prediction accuracy (Figure 2B, light gray bars) and performs worse under each game situation (early: 0.775±0.021, p<10−15, late: 0.621±0.018, p<10−17, scared ghosts: 0.672±0.025, p<10−12; two-sample t-test).

Secondly, we have revised Discussion to discuss the evidence supporting the hypothesis that monkeys used a hierarchical model.

“Our hierarchical model explains the monkeys’ joystick movements well (Figure 2B). Importantly, the strategies derived from the model can be verified with independent behavior measurements. The monkeys' fixation pattern, a measure of their attention, reflected the features associated with the current strategy (Figure 3E). Moreover, an increase of pupil dilation (Figure 3F), which was not associated with any particular changes of game states, was found at the deduced strategy switches. This is consistent with the prediction from the hierarchical model that there should be crucial decision-making of strategies around the strategy transitions.”

Finally, we have included a new paragraph in Discussion to point out that the cognitive constraint and planning complexity required in our Pac-Man task also suggest that the hierarchical model is more biologically plausible.

“In contrast to hierarchical models in which the decision-maker divides decision making into multiple levels and at each level focuses on an increasingly refined smaller set of game features (M. M. Botvinick et al., 2009; M. Botvinick and Weinstein, 2014; Dezfouli and Balleine, 2013; Ostlund et al., 2009; Sutton et al., 1999), a flat model’s decisions are directly computed at the primitive action level, and each action choice is evaluated with all game features. Although in theory a flat model may achieve equal or even greater performance than a hierarchical model, flat models are much more computationally costly. Especially when working memory has a limited capacity, as in the case of the real brain, hierarchical models can achieve a faster and more accurate performance (M. Botvinick and Weinstein, 2014). Our Pac-Man task contains an extensive feature space while requiring real-time decision-making that composes limitations on the cognitive resources. Even for a complex flat model such as Deep Q-Network, which evaluates primitive actions directly with a deep learning network structure without any temporally-extended higher-level decisions (Mnih et al., 2015), the game performance is much worse than a hierarchical model (Van Seijen et al., 2017). In fact, the most successful AI player to date uses a multi-agent solution, which is hierarchical in nature (Van Seijen et al., 2017). Our study shows that the monkeys also adopted a hierarchical solution for the Pac-Man game.”

3) The authors should acknowledge that monkeys do not always perform this task in the optimal way, and that monkeys perhaps use a more passive strategy, or use multiple strategies at the same time. The monkeys could obtain almost all the rewards by the end of the game as there is nothing to pressure or force them to optimize their choices. Authors should more fully account for this feature in the experimental design when interpreting their data.

1. The monkeys certainly did not always perform the task in the optimal way. Some sources of non-optimal behaviors are rather trivial. The joystick was not a natural instrument for the monkeys, and motor errors significantly contributed to their suboptimal behavior. Lapses of concentration were another factor. The monkeys might not have always played the game as attentively as we would like. The other sources of non-optimality, however, can be rather complicated. First, it is difficult to define optimality when we do not know what exactly the monkeys were optimizing. A lot of potential factors may have affected the monkeys’ decision making, including reward rate, temporal discounting, cost of effort (both mentally and physically), etc. What appears suboptimal may actually be optimal for the monkeys. The *suicide* behavior is a good example. Moreover, the computation of decisions in this task is complicated and limited by the monkeys’ cognitive resource. With an extensive state space, action sequences grow exponentially with the planning horizon, and the value of actions depends on the entire sequence of past actions, observations, and states (M. M. Botvinick et al., 2009; Kaelbling et al., 1998). The time constraint imposed by the task exacerbates this scaling problem and forces the monkeys to make decisions based on suboptimal heuristics.

2. The monkeys could not solve the task *passively*. The game is designed to force the players to make decisions quickly to clear the pellets, otherwise the ghosts would catch Pac-Man and end the game. Even in the monkey version of the game where the monkeys always get another chance, Pac-Man deaths lead to prolonged delays with no rewards. In addition, we provided additional rewards when a maze was cleared in fewer rounds (20 drops if in 1 to 3 rounds; 10 drops if in 4 to 5 rounds; and 5 drops if in more than 5 rounds), which added motivation for the monkeys to complete a game quickly.

There are additional clues from the monkeys’ behavior suggesting that the monkeys were actively making decisions to play the game. For example, both monkeys treated the two ghosts differently. They would sometimes follow Clyde when it was close (Figure 1C, solid yellow line, Pac-Man's moving tendency towards ghost Clyde was larger than 50%). As Clyde was programmed to move to the left corner of the map when Pac-Man was within 10 steps away, following Clyde was actually safe. Another good example is how the monkeys dealt with energizers and Pac-Man death. Our analyses of the *planned attack* and *suicide* behavior clearly demonstrated that the monkeys actively made plans to change the game into more desirable states. Such behavior cannot be explained with a passive foraging strategy.

In light of this discussion, we have revised the manuscript in several places to reflect the points above.

Reviewer #1 (Recommendations for the authors):1. What do you mean by saying "a reasonable level" (line 114)? You may want to show the criteria.

The criterium was determined subjectively. As shown in Appendix Figure 2D and 2J, monkeys’ behavior was stable within the period. We have collected more behavior data since the manuscript was written and submitted for review. But to keep consistency, the new data were not included in the revision. The new data are qualitatively very similar.

2. Although Clyde tends to keep a certain distance from Pac-Man, why does Pac-Man also chase it when they are close to each other? (figure 1C)

Just as in the original Pac-Man game, Clyde is programmed to move to the left corner of the map when Pac-Man is within 10 steps away. In addition, the ghosts do not move in the reversed direction. Therefore, it is actually safe for Pac-Man to follow Clyde when they are near each other.

3. Essential references are needed when explaining the behavior or mentioning previous literature, for example, in lines 316 and 485.

Done.

4. More details about how you tweaked the task are needed, such as how the monkeys operated the joystick? Holding it to move the Pac-Man, I guess.

Just as in the original Pac-Man game, the monkeys only need to flip the joystick toward a direction, and holding the joystick is not necessary. If the turn associated with the joystick movement is currently not possible, Pac-Man continues to move in the current direction until the turn becomes valid. Therefore, the monkey may move the joystick before the Pac-Man reaches a crossing to make a turn at that crossing.

Reviewer #2 (Recommendations for the authors):– I think there are potential problems with identifiability in the model as detailed. If, for instance, the authors used a model with known time-varying weights to play the game and then fit their model to these data, do they correctly recover the weights?

We have tested our model with simulated data generated with an artificial agent playing the game with time-varying strategy weights. The model could recover the ground-truth strategy label faithfully. The details of these analyses are included in the Supplementary (Appendix Figure 4).

– I am unfamiliar with the sense of "hierarchical" behavioral models as used here. By contrast, hierarchical models in RL involve goals, subgoals, etc., with models nested inside one another. If the authors are borrowing this less common sense of the term from somewhere, it would be helpful to cite it. I am not sure in what sense a perceptron is a "flat" model.

In the new revision, we define the flat model in Results as follows:

“… In contrast, in a flat model, decisions are computed directly for the primitive actions based on all relevant game features…”

In Discussion, we further point out:

“In contrast to hierarchical models in which the decision-maker divides decision making into multiple levels and at each level focuses on an increasingly refined smaller set of game features (M. M. Botvinick et al., 2009; M. Botvinick and Weinstein, 2014; Dezfouli and Balleine, 2013; Ostlund et al., 2009; Sutton et al., 1999), a flat model’s decisions are directly computed at the primitive action level, and each action choice is evaluated with all game features…”

We further revised the manuscript in multiple places to discuss how our results support the hierarchical decision-making by the monkeys.

– Figure 1B: I find it hard to distinguish the grayscale lines and match them with their corresponding choice types. Also, I don't see standard errors here, as indicated in the caption.

Fixed.

– Figure 2: I found the boxes around the figure panels visually unappealing. This occurs in a couple of other figures.

We have fixed the problem of pdf conversion and hopefully the figure quality issue has been solved.

– Figure 2C-D: If I follow, these are violin plots, not histograms, as stated in the caption.

Fixed.

– Figure 2E: The authors might consider a radar chart as an alternative way of representing these proportion data. Such a chart would allow them to plot each column here as one line and might facilitate more direct comparisons across strategy proportions.

We tried Radar chart (Author response image 1), but there are too many overlaps between the four game contexts, making the figure hard to read. Therefore, we have kept the bar chart.

**Author response image 1. sa2fig1:** Radar chart of the labeled dominating strategy ratios across four game contexts.

– ll. 206-210: Before having read the methods, this text does not make it clear to me what the authors have done. That is, I don't know what it means here to linearly combine basis strategies or to "pool predictions." More specific language would be helpful, even if just a few words.

We have revised the corresponding section to clarify the fitting process:

“At any time during the game, monkeys could adopt one or a mixture of multiple strategies for decision making. We assumed that the final decision for Pac-Man’s moving direction was based on a linear combination of the basis strategies, and the relative strategy weights were stable for a certain period. We adopted a softmax policy to linearly combine utility values under each basis strategy, with the strategy weights as model parameters. To avoid potential overfitting, we designed a two-pass fitting procedure to divide each game trial into segments and performed maximum likelihood estimation (MLE) to estimate the model parameters with the monkeys’ behavior within each time segment (See Methods for details). When tested with simulated data, the fitting procedure recovers the ground-truth weights used to generate the data (Appendix 1—figure 4).”

– Figure 3C-D: I don't think the dotted magenta lines are necessary, and they make the numbers hard to read.

Fixed.

– I was not sure what to take away from Figure 4. As a conceptual description, perhaps it belongs earlier in the text? Perhaps with some schematic as to how models are combined?

Figure 4 introduces the compound strategies and leads to our analyses of compound strategies. With Figure 4, we would like to wrap up all crucial concepts developed in the above texts and emphasize the relationships between task, compound strategy, basis strategy, and primitive action selections. We played with the idea of having a schematic illustration earlier in the text but were worried that readers might not sufficiently appreciate the illustration without the relevant analyses. We certainly are still open to adjusting the writing flow should the reviewers and the editors favor the alternative.

– ll 748-749: Shouldn't utilities for impossible actions be -∞? That is, in any probabilistic action selection model, there will be a probability of selecting any action with a finite utility difference, so the difference would need to be infinite to rule out an action.

Thanks for the suggestion. We have now adopted this suggestion, re-done our analyses, and updated all figures under this modification. The results were qualitatively similar to the original.

– ll. 918-920: Three standard deviations seems like a pretty drastic cutoff for outliers. Why not five or at least four? Does this make a difference?

We tried five and four as alternative cutoffs. The results are very much the same. So we did not change the original plot.

**Author response image 2. sa2fig2:** The monkeys’ pupil diameters around the strategy transition under different cutoff conditions.